# Thermo-adaptive interfacial solar evaporation enhanced by dynamic water gating

Yi Wang[1,2,3] ✉, Weinan Zhao[1,3], Yebin Lee[1], Yuning Li [1], Zuankai Wang [2] & Kam Chiu Tam [1] ✉

Solar-driven evaporation offers a sustainable solution for water purification, but efficiency losses due to heat dissipation and fouling limit its scalability. Herein, we present a bilayer-structured solar evaporator (*SDWE*) with dynamic fluidic flow mechanism, designed to ensure a thin water supply and self-cleaning capability. The porous polydopamine (*PDA*) layer on a nickel skeleton provides photothermal functionality and water microchannels, while the thermo-responsive sporopollenin layer on the bottom acts as a switchable water gate. Using confocal laser microscopy and micro-CT, we demonstrate that this unique structure ensures a steady supply of thin water layers, enhancing evaporation by minimizing latent heat at high temperatures. Additionally, the system initiates a self-cleaning process through bulk water convection when temperature drops due to salt accumulation, thus maintaining increased evaporation efficiency. Therefore, the optimized *p-SDWE* sample achieved a high evaporation rate of 3.58 kg m$^{-2}$ h$^{-1}$ using 93.9% solar energy from 1 sun irradiation, and produces 18–22 liters of purified water per square meter of *SDWE* per day from brine water. This dynamic water transport mechanism surpasses traditional day-night cycles, offering inherent thermal adaptability for continuous, high-efficiency evaporation.

Freshwater scarcity is becoming a global concern, necessitating the development of effective water collection and purification technologies. More recently, interfacial solar-driven evaporation has emerged as a sustainable method for generating clean water using solar energy. Additionally, it acts as a complementary platform coupled with other solar-driven technologies such as photocatalysis and desalination, enriching the spectrum of sustainable water treatment solutions[1].

Recent advancements in solar evaporators have been driven by the development of materials with high photothermal efficiency[2], such as metal nanoparticles[3], carbonaceous materials, semiconductors, and polymers. These advancements have led to significant improvements in evaporation efficiency, achieved by innovatively integrating evaporators with varied structural forms including 3D porous architectures[4], 2D lamellae[5], and 1D columnar structures. Furthermore, enhancements in evaporation rates have been facilitated by a range of strategies, encompassing the rational design of water channels[6], effective water supply control[7], and the implementation of multistage evaporation processes[8]. Scaling this technology for seawater treatment has encountered challenges, primarily by salt fouling in the water channels. Innovative strategies such as Janus structures[9], salt-rejecting structures[10], contactless design[11], and localized crystallization[12,13] have been deployed to mitigate this issue. Despite their efficiency and reduced salt fouling, current systems are hindered by their rigid structural designs and passive operational cycles, impeding their capacity for sustained operation. Thus, attaining autonomous solar-driven water evaporation continues to be a formidable challenge.

[1]Department of Chemical Engineering, Waterloo Institute for Nanotechnology, University of Waterloo, 200 University Avenue West, Waterloo, Ontario N2L 3G1, Canada. [2]Department of Mechanical Engineering, The Hong Kong Polytechnic University, Hong Kong, China. [3]These authors contributed equally: Yi Wang, Weinan Zhao. ✉e-mail: wangiiupc93@gmail.com; mkctam@uwaterloo.ca

Smart materials are catalyzing a transformative shift in robotics, paving the way for autonomous soft robots capable of self-adapting to environmental stimuli like heat, light, and magnetism[14]. However, leveraging this inherent self-adaptability in solar evaporator is still nascent. A key challenge lies in advancing solar evaporators is improving their responsive precision to enable self-regulation, thereby maintaining both a high performance and stable operation autonomously, without the need for external interventions. While several and pioneering studies, like magnetic[15] and ammonia-responsive[16] evaporators, have been reported, these evaporators struggle with formidable limitations, such as strict magnetic field requirement, and loss functions after post-contamination. Notably, thermo-responsive materials, with their lower critical solution temperature (LCST) behavior, hold the potential to harness temperature fluctuations induced by fouling or salt accumulation during the solar-driven evaporation process. These microscale LCST shift profoundly impact the near-surface wettability and water structure variation through microparticle grafting, as reported in our previous studies with sporopollenin[17]. The integration of these materials into solar evaporator domain will not only offer promising heightened sensitivity to environmental shifts but also ensures unmatched stability in system performance.

Here, we report a bilayer-structured solar evaporator featuring a dynamic water-thermal controlling system, autonomously shifts between efficient thin water evaporation and salt washing. Unlike previously reported solar evaporator with rigid structures, our *SDWE* features a switchable water transport channel that adapts to temperature changes during evaporation, enabling improved evaporation efficiency continuously, even in high-concentration saline brine. We fabricated *SDWEs* using nickel foam as the substrate and incorporated two key components: the interfacial polydopamine nanosphere-assembled layer and the bottom thermo-responsive sporopollenin-engineered layer (*PNm-g-SEC*). Specifically, the upper *PDA* layer serves as the photothermal interface, while the lower layer, made of thermo-responsive sporopollenin, acts as a switchable gating layer. At higher temperature, this layer transitions to a superhydrophobic state at elevated temperatures, directing water through specific *PDA*-assembled channels and facilitating a consistent supply of thin water layers, optimizing the evaporation process. At lower temperatures, the *PNm-g-SEC* layer becomes hydrophilic, drawing bulk water to backflow accumulated salt. This dual-action mechanism not only ensures continuous water transport but also self-regulates to address salt accumulation. Moreover, supported by the sporopollenin's hollow structure that provides thermal insulation, this unique *SDWE* design yielded a high solar-vapor conversion performance, with a vapor generation rate of up to 3.58 kg m$^{-2}$ h$^{-1}$ and 93.9% solar-to-vapor efficiency under 1 sun irradiation.

## Results

### Design of the solar-driven water evaporator

Figure 1 illustrates the construction units and working principles of the designed solar water evaporator. Our solar evaporator, namely *SDWEs*, comprises two essential components modified on nickel foam, namely the interfacial *PDA* layer and the bottom *PNm-g-SEC* layer. The bilayer structures work synergistically to enhance the solar water generation performance by balancing the trade-offs between water transport rate and light-to-heat energy consumption during the water phase-transition evaporation process. Specifically, the introduction of porous *PDA* coating layer along the skeleton enabled the continuous water supply of controllable and varying amounts. By incorporating the thermo-responsive pollen microparticles on the bottom side in contact with seawater, *PNm-g-SEC* particles performed as a solid gate to guide the thin water passing through the hydrophilic microchannel that effectively block the flow of large amounts of bulk water through the external microchannels originating from the pore of the foam. This

switch-like mechanism orchestrates a selective and directional supply of thin water layers at elevated temperatures during evaporation. Concurrently, it facilitates an autonomous shift to bulk water self-washing when surface temperatures are influenced by pollutants or the natural day-night cycle, ensuring optimal operation under varying conditions. Moreover, the distinctive textures originating from the bilayer design rationalize the thermal energy management. The low thermal diffusivity of *PDA* combined with the porous topographical texture enhance the light absorption, while the hollow microcapsule structure of SEC with trapped air pockets serves as an effective heat resistance layer to reduce the thermal dissipation. Thus, when exposed to sunlight, the encapsulation black layer could efficiently achieve an optimal photothermal conversion efficiency, raising the temperature gradient and maintaining a stable high temperature within the system to further improve the evaporation performance.

The fabrication of *p-SDWE* foam was conducted in two steps, as schematically shown in Fig. 2a. First, *p-PDA* nanospheres were first deposited onto the porous nickel foam (*NiF*) with pore size ($\approx 200$ μm) using in-situ polymerization method[18]. The color of *NiF* transitioned from silver gray to black after being modified with an adherent *p-PDA* layer, which simultaneously endowed the foam with a remarkable light absorption characteristic. The *PDA* coating layer on the substrate served as a thin water transport layer, while the top section of the evaporator also functioned as the photothermal layer. Subsequently, the incorporation of thermo-responsive sporopollenin onto the bottom section of evaporator was achieved by a facile di*p*-coating process[19]. The *p-PDA@NiF* was dipped into *PNm-g-SEC* solution at a certain immerse depth, where the bottom skeleton of the foam coated with *PNm-g-SEC* turned into dark brown after vacuum drying. Owing to the unique temperature-dependent water affinity, *PNm-g-SEC* decorated at the bottom section served as the water transport gating layer to enable the selective thin water supply.

Specifically, monodispersed *PDA* spheres grew on the skeleton of the foam in water-ethanol-ammonia mixed solvents at room temperature (Fig. 2b). By adjusting the water-ethanol solvent reactive conditions, various morphologies of *PDA* layers were produced according to the Hansen solubility parameters[20], designated as smooth *PDA* layer (s-*PDA*), roughness *PDA* layer (r-*PDA*) and porous *PDA* layer (*p-PDA*), respectively. The *p-PDA* nanospheres, with a diameter of 520 nm, assembled into micro clusters and assembled into interconnected channels as demonstrated in scanning electron microscopic image (SEM) of Fig. 2c, which introduced onto the skeleton with porous structures and facilitated water transport. s-*PDA* samples featured with the flat topography and r-*PDA* samples displayed a continuous bumpy microstructure with large *PDA* clusters (Supplementary Fig. 1).

Moreover, the water transport gating layer was fabricated by coating *PNm-g-SEC* onto the foam skeleton as shown by the SEM images of Fig. 2d, e. The *PNm-g-SEC* was prepared by grafting sunflower sporopollenin with N-isopropylacrylamide (NIPAM) via free radical polymerization under nitrogen environment. SEM images of Fig. 3a revealed the morphology of *PN10-g-SEC*, exhibiting triangular shapes decorated with microridges, and the polymer-grafted canopy on *PN10-g-SEC* particles featured with high roughness nanostructure. Additionally, the particle size analysis confirmed the successful preparation of *PN10-g-SEC* particles with a uniform size distribution of ~26.9 μm (Supplementary Fig. 2). As shown in Fig. 3b, the electron EDS elemental mapping images showed that the C, N, and O elements were uniformly distributed within the *PN10-g-SEC* microparticles. Successful PNIPAM-grafted layer was further characterized by X-ray photoelectron spectroscopic (XPS) elemental analyzes, demonstrating the presence of organic molecules with the expected changes in the ratio of N−C bond on the surface of *PN10-g-SEC*[21] (Supplementary Fig. 3a and Supplementary Table 1). Besides, Fourier transform infrared (FTIR) spectra confirmed the presence of amino groups, where the peak at 1650 cm$^{-1}$

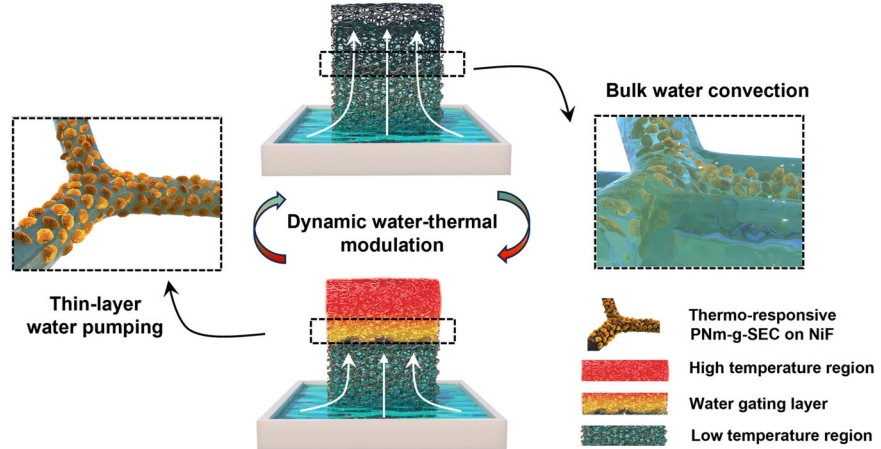

**Fig. 1 | Illustration of the working principles of *SDWE*s.** Water transport occurs along the larger pores within the nickel foam at low temperature (right), while thin water transport takes place along the intine *PDA* layer at high temperature (left), driven by the thermodynamic modulation induced by the water gating layer of *PNm-g-SEC*.

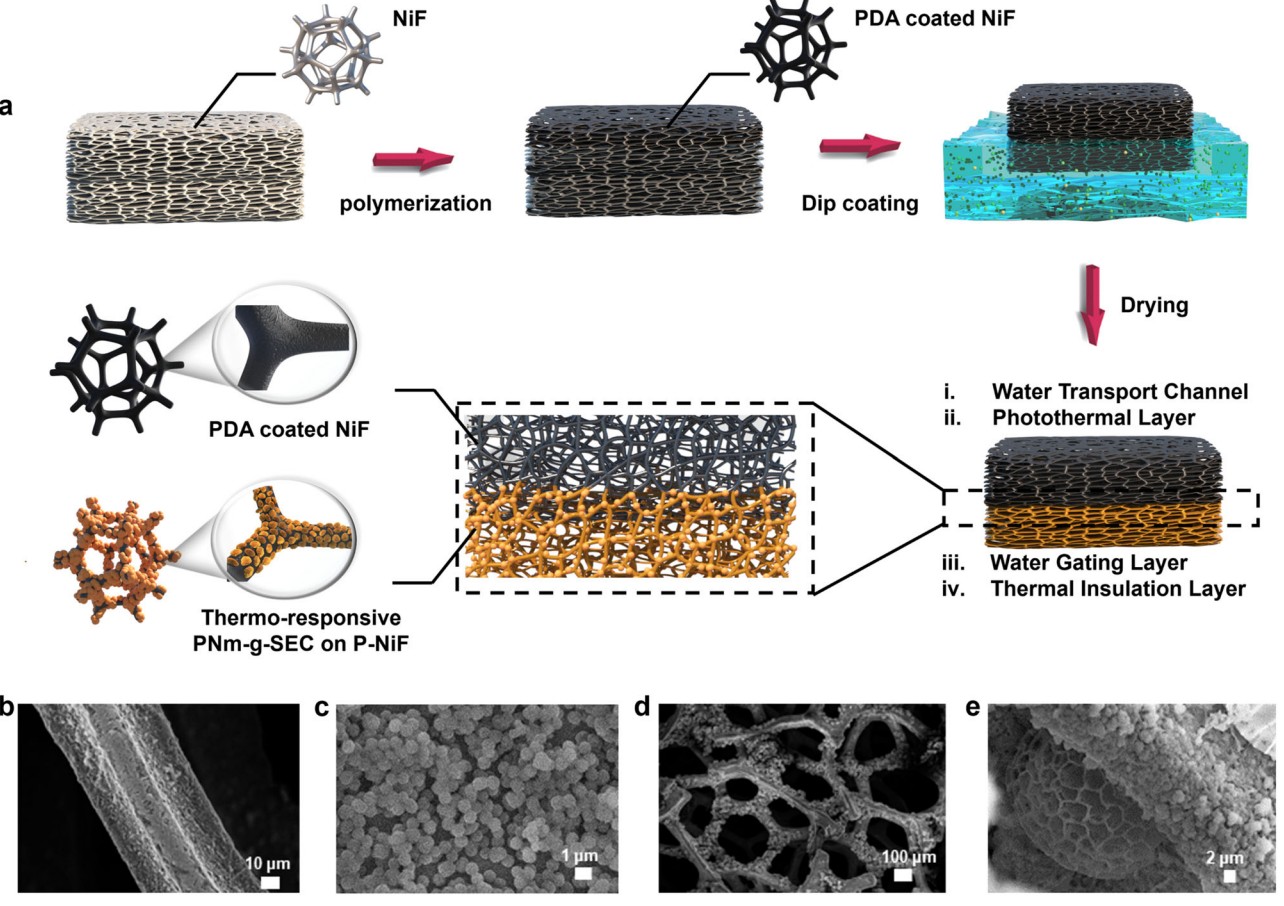

**Fig. 2 | Design of bilayer-structured solar evaporator. a** Schematic illustration of the fabrication procedure of bilayer-structured evaporator, where porous *PDA* photothermal layer was fabricated via in-situ polymerization and following by a facile di*p*-coating process, the thermo-responsive sporopollenin layer was assembled on the bottom skeleton of *NiF* foam. SEM images of *p-PDA* coating layer. **b** The porous structure of foam skeleton. **c** *p-PDA* nanospheres, SEM images of water transport gating layer **d** The skeleton decorated with *PNm-g-SEC* micro-particles. **e** The interaction between *PNm-g-SEC* microparticles and *PDA* layer.

is associated with the N−C=O bond and the absorption peak at 1550 cm$^{-1}$ corresponds to the N−H bonds[22] (Supplementary Fig. 4). In addition to the thermal responsive characteristics, the polymer conformational transition of *PNm-SEC* was confirmed by UV-vis spectro-photometry and Fig. 3c displays the LCST of *PNm-g-SEC* with a LCST of 33.2 °C. As shown in Fig. 3d, the broad Raman band of *PN10-g-SEC* extending from 3000 cm$^{-1}$ to 4000 cm$^{-1}$ indicated the vibration of

hydrogen bonds in water. Within this range, the 3250 cm$^{-1}$ peak cor-responded to free water with four hydrogen bonds and -$CH_3$ groups surrounded by polyhedral cages composed of tetrahedrally hydrogen-bonded water molecules (blue-marked region), which is known as 4-coordinate hydrogen-bonded water (4-HBW). The peak at 3410 cm$^{-1}$ represents the in-phase vibrations of water molecules captured by the C=O or N−H groups (yellow-marked region), which is referred to as

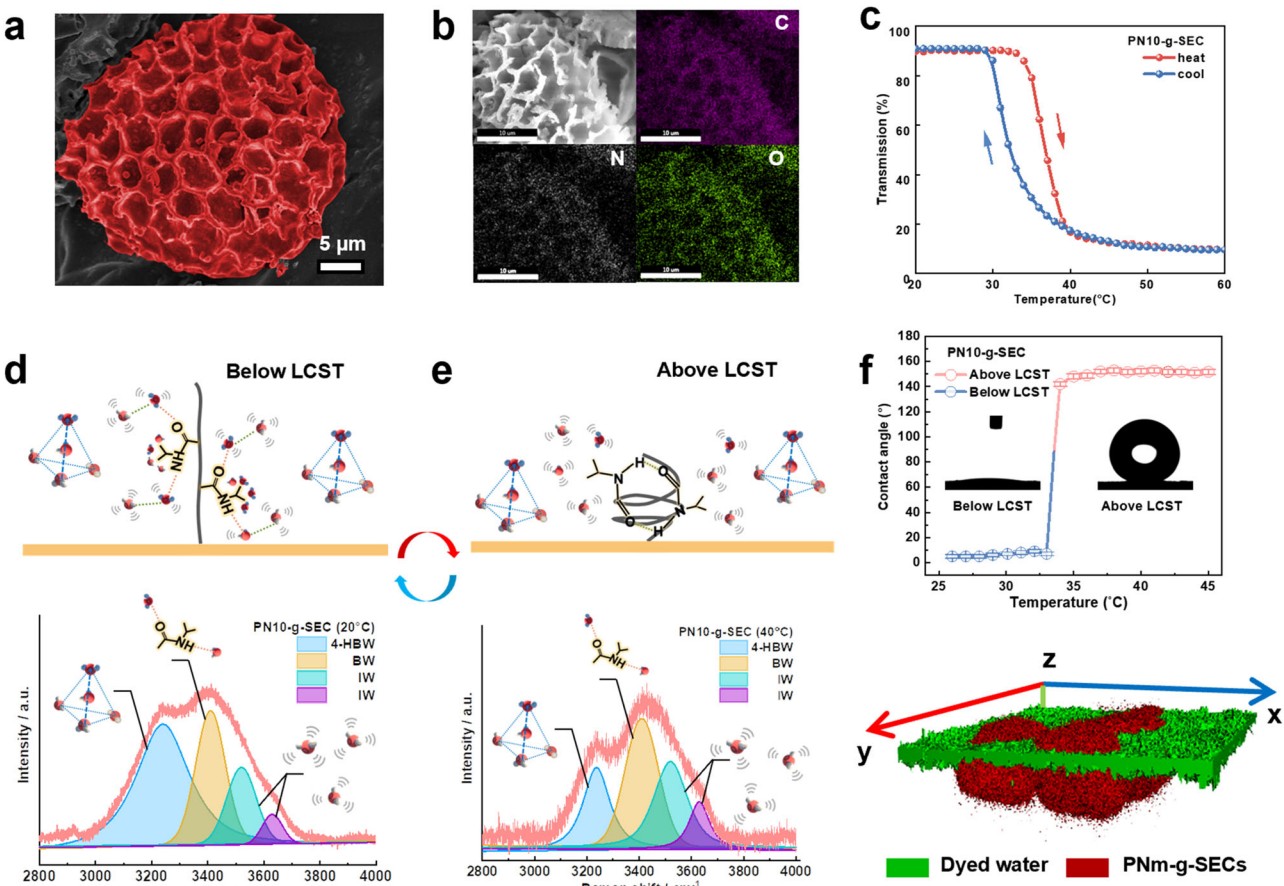

**Fig. 3 | Surface properties of evaporators modified with *PNm-SECs*. a** SEM images of *PN10-g-SEC* microparticles. **b** EDS mapping images of *PN10-g-SEC* microparticles. **c** LCST behaviors of *PN10-g-SEC* confirmed by UV-vis spectrophotometry, Interfacial water structure along polymer-grafted layer of *PN10-g-SEC* measured by Raman spectroscopy: (**d**) below LCST (**e**) above LCST. **f** Water contact angle

changes driven by LCST, where the bottom confocal laser microscopic images illustrating the contact line between pollen surface-air-water above LCST. 4-HBW: free water with four hydrogen bonds, BW: tetrahedrally hydrogen-bonded water molecules, IW: intermediate water, LCST: lower critical solution temperature.

bound water (BW). Besides, the peaks at 3520 and 3630 cm$^{-1}$ correspond to the stretching of weakly or non-hydrogen-bonded water molecules (green and purple-marked regions), known as intermediate water (IW), indicating the presence of hydrophobic disordered water in the hydration shell[23]. At 20 °C, the higher ratio suggested that the PN10-g-L.SEC surface had a larger amounts of bound water, with the hydrophilic C=O and -NH$_2$ groups predominantly positioned at the air/*PNm-g-L.SEC*/water interface. Conversely, the spectrum displayed a dramatic reduction in the 4-HBW/BW ratio at 40 °C, indicating that the water underwent a transformation into a less ordered and weaker hydrogen-bonded structure, while the hydrophobic -CH$_3$ groups rearranged and exposed at the interface (Fig. 3e). Taking advantage of this characteristic together with the hierarchical structure on the outer surface of SEC, we obtained the thermo-responsive surfaces with switchable wettability transition as illustrated in our previous study[17]. By adjusting the polymer grafting density, we successfully obtained pollen microparticles with hydrophilic/hydrophobic transition states displayed on *PN20-g-SEC* surfaces (Supplementary Fig. 5) and the constructed *PN10-g-SEC* surfaces exhibited remarkable superhydrophobic/superhydrophilic wettability transition characteristics. The water contact angles (θ$_w$) observed on the surface assembled with *PN10-g-SEC* particles underwent a transition, switching from 2° at 20 °C to 153° at 35 °C, clearly demonstrating the potential for effective water management (Fig. 3f). To further confirm the water-solid surface interaction, we imaged the 3D interface in contact with the liquid droplet and integrated *PN10-g-SEC* surface using the confocal laser

microscopy at different temperatures[24]. The green section (fluorescence-stained) is represented as the water phase in the confocal images, while the red section is associated with *PN10-g-SEC* microparticles (Fig. 3f). At the temperature below the LCST, the green section was observed from the base substrate to the outer surface, illustrating the droplet tendency to fully wet the micro-particle surface and fill the gap between the microparticles. In contrast, if the droplet came into contact with the *PN10-g-SEC* surface at a higher temperature (above LCST), we could observe the liquid baseline suspended between particles, indicating a non-wetted state. This result demonstrated that the temperature could readily switch the droplet-surface interaction and drive the water transport along the preferred channel. This thermo-responsive behavior combined with near-surface water phase regulation is the key to achieving a dynamic water control in the solar evaporation process, which will be discussed in the following section.

## Photothermal effect and thermal localization

Under constant solar illumination (1 kW m$^{-2}$), solar vapor generation performance of the *SDWEs* (5 × 5 cm area, 1.0 cm thickness) was recorded by mass change of evaporated water and temperature distribution of *SDWEs* with time evolution (Supplementary Fig. 6). It is obvious that the vapor generation using the *SDWEs* was more efficient compared to pure water evaporation without *SDWEs* under identical illumination conditions. The water evaporation rates were calculated from the slope of mass change curves. Under 1 sun illumination, *p-SDWE* achieved the highest rate up to 3.58 kg m$^{-2}$ h$^{-1}$ among all

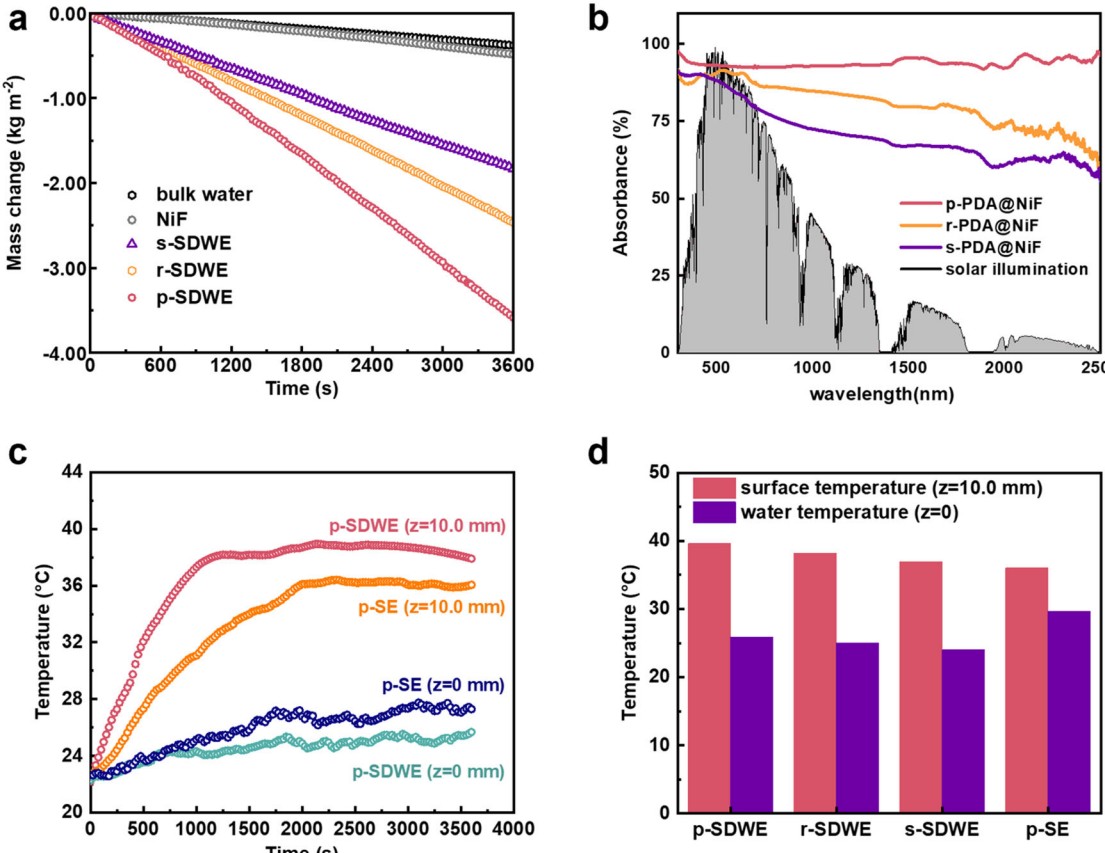

**Fig. 4 | Solar evaporation performance and thermal localization of *SDWE*s.**
**a** Mass change of water over time of the *SDWE* foams under one sun solar illumination. **b** UV-vis-NIR diffuse reflection spectra of the *PDA*-coated *NiF* in the wavelength ranged from 200 to 2500 nm. **c** Temperature variations in *p-SDWE* SDWEs samples, which was 1.3, 2.2 and 5.2 times that for *s-SDWE*, *r-SDWE* and pure water, respectively (Fig. 4a). These results revealed that *p-SDWE* has an optimized water transport channel and economical energy consumption.

compared to the reference sample *p-DF* without decorating the SEC layer, with *z* denoting distance from the top water surface. **d** Temperature gradient of *SDWE* foams characterized by infrared thermal images under one sun solar irradiation.

An efficient photothermal layer together with a heat insulation layer enabled the localized surface heating on the evaporator while alleviating unexpected heat conduction to the surrounding environment, which is one key factor for improving the solar evaporation rate. Fig. 4b shows the UV-visible–near-infrared (UV-vis-NIR) absorbance spectra of the *PDA@NiF* foams, with the *p-PDA* achieving an efficient and broadband solar absorption (~93.6%) over a wide wavelength range (200–2500 nm). This exceptional performance can be attributed to the presence of the indole-5,6-quinone unit within the *PDA*, which provides ample conjugated structures for extensive light absorption. The electron donor-acceptor dynamics between the 5,6-dihydroxyindole and indole-5,6-quinone units effectively narrow the energy bandgap, facilitating electron transfer. Complementing this, the material's inherent darkness, highlighted in Supplementary Fig. 3, confirmed its robust absorption in both visible and NIR spectrums. Additionally, by strategically engineering the *PDA* nanoparticles' assembly, we enabled the photothermal layer with low light reflectance of *p-PDA@NiF* foams of ~5.6%. This optimization contrasts with the less efficient sunlight absorption observed in *s-PDA* and *r-PDA* (Supplementary Fig. 7a). The densely arranged nanospheres within the interconnecting pores of the *NiF* structure contribute to this efficiency, capturing solar light and minimizing reflectance. This effect is further amplified by the foam's hierarchical porosity, which promotes intense diffuse reflection[25]. The dry surface of the *SDWE* rapidly heats to nearly

63 °C, as evidenced by the temperature profiles in Supplementary Fig. 7b, showcasing its remarkable solar-to-heat conversion performance. During the evaporation process, which inherently involved a cooling effect, the energy transformation into heat was further confirmed through IR thermal images and temperature profiles (Fig. 4c). The surface temperature of *p-PDA* increased rapidly and reached a plateau at around 39.6 °C, while the temperature of bulk water only increased slightly. Furthermore, the surface temperature of *p-SDWE* displayed a higher temperature increase compared to the solar evaporator with an *s-PDA* layer (38.1 °C) and an *r-PDA* layer (36.9 °C) under identical light irradiation condition (Fig. 4d). This distinct temperature elevation demonstrates the *p-SDWE* system's proficiency in harnessing and utilizing solar energy for efficient water evaporation.

High solar-to-thermal conversion coupled with efficient heat transfer is crucial for rapid water evaporation. Considering the increase in topographical texture with effective phonon scattering and low thermal diffusivity in *PDA* (0.08 W m⁻¹ K⁻¹), the thermal energy dissipation must be much slower than the accumulation. The localized absorption of heat at the PDA water interface optimizes the phase change from water to vapor. Differential scanning calorimetry experiments revealed a lower vaporization enthalpy for water in *PDA*-assembled microchannels compared to bulk water (Supplementary Method 1, Supplementary Fig. 8 and Supplementary Table 1). To investigate the variation in hydrogen bonding between the confined thin water layer on *PDA* surface and bulk water, we analyzed the hydrogen bond vibrations by Raman spectra (Supplementary Fig. 9). The increase in intermediate water (IW) suggests that the stable hydrogen bonding network of bulk water was disrupted on the *PDA*

surface. This alteration facilitates the release of small water clusters during evaporation, aligning with the observed reduction in vaporization enthalpy[26]. Besides, the lithium-ion solvation experiment supports this analysis, indicating a tendency for water cluster formation during evaporation (Supplementary Method 2 and Supplementary Fig. 10). These findings imply that the existence of a thin water layer, enhanced by the surface characteristics of polymeric materials such as PDA, plays a significant role in boosting evaporation efficiency.

Additionally, maintaining heat at the evaporator layer while preventing undesirable heat dissipation, such as that caused by convective water flow, necessitates an effective heat insulation layer. SEC, with its hollow structure encapsulating air pockets, emerges as an excellent candidate for a heat resistance layer (Supplementary Fig. 11). This design capitalizes on the low thermal conductivity of air, enhancing the material's insulating properties[27]. From the results in Fig. 4c and Supplementary Fig. 12, in the thermal steady state, the surface temperature was more than 13.7 °C higher than the bottom water temperature (25.9 °C). In contrast, the solar evaporator (p-SE) acted as a controlled sample by removing the SEC layer, resulting in a lower surface temperature increase of approximately 36.2 °C and ultimately raising the bottom water temperature to 27.4 °C. This was attributed to the strong thermal localization of hollow PN10-g-SEC layer, which effectively reduced excess convective heat losses to the water beneath and further enhanced the evaporation efficiency. Therefore, the introduction of PN10-g-SEC layer within the SDWEs could maintain the lower temperature fluctuations and reduced the heat loss, revealing the potential improvement of evaporation rate for outdoor operation.

## Dynamic water control for high-efficiency solar water evaporation

Initially, we investigated the temperature-responsive, switchable water transport in SDWEs, delving into microscale water flow dynamics. As discussed previously, the PN10-g-SEC layer featuring hydrophilic at lower temperatures, is able to pump bulk water vertically to the surface and thus fill the NiF's larger pores. Upon exceeding its LCST, the PN10-g-SEC layer undergoes a transformation to a water repellent state that impedes water from traversing the inner pores, thereby promoting a consistent supply of a thin water layer (Fig. 5a, b). To illustrate this concept, we introduced fluorescent-dyed into the foams and tracked their motion using time series-confocal microscopy. When injecting water below the LCST of 25 °C, it swiftly ascends through the NiF framework, leading to rapid and complete saturation of its inner pores. Fig. 5c and Supplementary Movie 2 illustrated this process, with the water's complete infiltration into the nickel foam clearly marked by a fluorescent blue signal. Conversely, as visualized in Fig. 5d and Supplementary Movie 1, the water at 36 °C preferential adheres to the skeleton of the p-SDWE and flow rapidly along the entire framework. Additionally, the water layer within the top layer of PDA@NiF coating was determined to have a thickness of 5–8 μm. The results illustrated that porous PDA layer have strong water affinity and capillary force, which could achieve directional ultrathin water transport.

Furthermore, the ability of our SDWEs to regulate the continuous water supply rate is essential for maintaining efficient vapor generation. We designed porous PDA coating layers on the skeleton of NiF as water wicking layer that produced narrow water transport channels within the interconnected pores of NiF and the pore generated by the PDA nanospheres (Fig. 5b). Notably, the water transport rate is affected by different factors, such as the topography, and the water affinity of the channel. We first investigated the influence of topography of PDA layer on the water transport rates of three types of coating surfaces (s-PDA, r-PDA, p-PDA). To simplify the water management configurations, three different PDA-coated nickel plates were used to compare the vertical water transport rates. These plates were positioned upright in a water-filled environment, and the changes in the water height were captured using a high-speed camera. As shown in Supplementary

Fig. 13a, water could be rapidly transported upward on the p-PDA surface with a water transport rate of up to 10.0 mm s⁻¹. In contrast, the s-PDA and r-PDA surfaces, the rate of water transport was limited to 0.7 mm s⁻¹ and 2.10 mm s⁻¹ respectively. While water cannot be transported upward on the plate without the PDA layer, it could only attain a certain height associated with the capillary action[5]. Additionally, the water transport rate was independent of the thickness of the PDA layer, and it had an impact on the evaporation rate (Supplementary Fig. 13b and 13c), where thicker water layers displayed lower evaporation efficiency, while ultrathin water layers led to salt accumulation[28]. This phenomenon was induced by distinct capillary force generated during the water pumping process, where water pumping on p-PDA surface utilized the roughness structure of the wall and internal capillary force.

In addition to the topography and diameter of the channel, the water affinity also played an important role in the capillary force that was reflected by the water transport rates[29]. The wettability characteristics of the PDA surface were key factors controlling the interaction of water and the surfaces that were described by the water contact angle, surface free energy and adhesive force. Supplementary Fig. 14a shows that the water contact angle of p-PDA surface was 4.6° and was superhydrophilic, where water could easily spread (1 s) and penetrated the coating layer in 2.5 s. r-PDA possessed higher hydrophilicity with a water contact angle (CA) of 12.5 ± 1° compared to s-PDA layer with a CA of 32.1 ± 1°. In addition, the surface free energies calculated based on Owens, Wendt, Rabel and Kaelble (OWRK) theory demonstrated the surface attractive force towards polar/non-polar surfaces[30]. Specifically, p-PDA surface with 72.3 mJ m⁻² possessed a higher affinity towards water compared to the other two surfaces, where the r-PDA and s-PDA surfaces possessed a value of 70.1 mJ m⁻² and 67.6 mJ m⁻², respectively (Supplementary Note 1).

Simultaneously, to assess water transport on a macroscale, we simulated real-world conditions encountered during the solar-driven water evaporation. This involved characterizing two key processes: the supply of a thin water layer during vapor generation and the bulk water backflow during salt washing. We initially characterized the selective water transport characteristic induced by PN10-g-SEC layer during the solar steam generation, by recording the pumped water states using an optical microscope. The p-SDWE were placed onto the dyed water, and the water transport upwards within the pore and along the skeleton was recorded. Water easily pumped up via the capillary force generated by the pores of the Ni foam, where the dyed water filled the foam at lower temperature (Fig. 6a). In contrast, under higher temperature of 36 °C, a thin water layer along the skeleton of p-SDWE was observed on the topside but it did not block the pores of the evaporator layer (Fig. 6b). Therefore, these temperature-dependent wettability transition characteristics endow the PN10-g-SEC with the ability to act as a smart gate, enabling the modulation of the water transport channel within the foam pores during the evaporation process. Thin water is capable of being pumped and continuously supplied along the intine PDA water transport microchannel under solar illumination.

Conversely, bulk water was directed to the topside during the night when solar illumination was absent, subsequently facilitating the removal of salt crystals. To intuitively showcase the salt backflow capacity, we investigated the salt dissolution process by introducing 1 g of sodium chloride to the evaporator surface during vapor generation[31,32]. The salt progressively dissolved, completely disappeared within two hours as shown in Supplementary Fig. 15a. Additionally, the bulk water transport to the top surface of the evaporator rapidly back flowed saturated brine to bottom water container under the action of a salinity gradient and gravity, as recorded in Supplementary Fig. 15b and Supplementary Movie 3. These observations confirmed the effectiveness of temperature-induced bulk water flow in removing salt, ensuring that the rate of salt backflow from the evaporator surpassed its rate of salt generation, thus ensuring the sustained utilization of SDWEs.

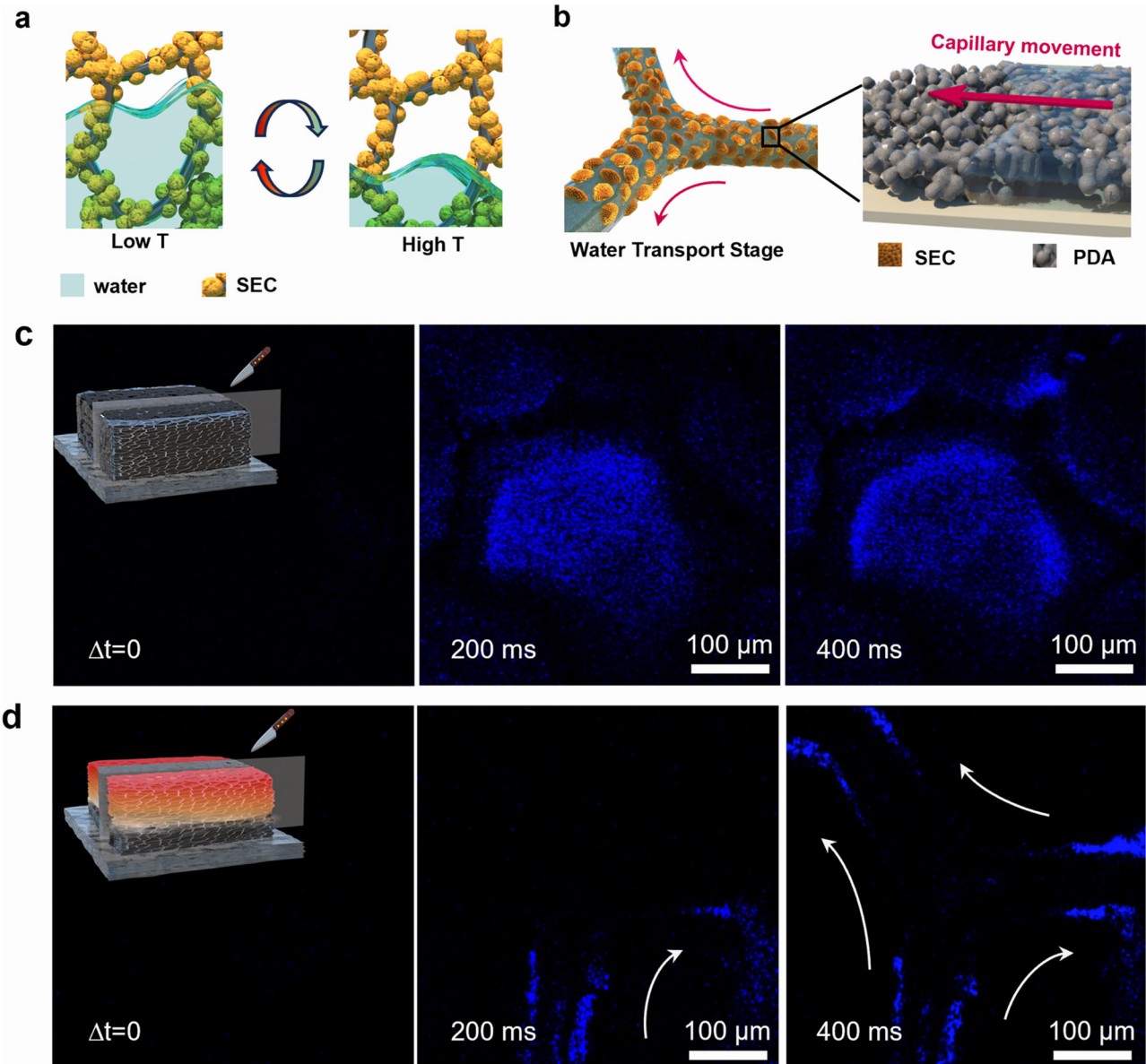

**Fig. 5 | Dynamic water flow engineered in the evaporators. a** Schematic illustration of water transport within switchable channel of *SDWE*s. **b** Thin water supply within *PDA*-assembled microchannel driven by capillary force, Confocal microscopic images of switchable water transport along *p-SDWE* using a time–series model at the *z*-axis: **c** bulk water fill the inner large pore of *NiF* foam, where the injecting water phase at 20 °C. **d** thin water transport along the *PDA*-assembled channels on the skeleton of *NiF* foam, where the injecting water phase at 36 °C. Δt: Time interval, Low T: low temperature, High T: high temperature.

To examine the evaporation efficiency for the whole process, we recorded the water loss under transient conditions, enabling the comparison of the evaporation rates. The mass change of water within the *SDWE* was observed to possess three distinct stages, as depicted in Fig. 6c. The initial stage involved the light-to-heat conversion and transfer, where natural water evaporation occurred[33]. Following this process, as the bulk water filled the pores, evaporation occurred throughout the entire foam at a rate of 1.20 kg m$^{-2}$ h$^{-1}$. Since the capillary force generated by the pores of the foam itself is one of the driving forces for pumping water from bottom to the top surface, flooding of water in the foam is inevitable. In this situation, the solar energy is wasted to heat the excess bulk water, resulting in lower solar-driven water evaporation rates. These two stages are referred to as the response period of *SDWE*s over a time period of around 600 s, where the evaporation rate was typically calculated after this unstable state in many previous studies. Noteworthily, upon transition to the third stage, characterized by the infusion of a thin water layer, the foam's

evaporation rate markedly increased to 3.81 kg m$^{-2}$ h$^{-1}$. Upon solar irradiation, the foam's surface temperature rose above the LCST to 39.6 °C, prompting the thermo-responsive *PN10-g-SEC* microparticles to exhibit hydrophobic properties. Consequently, these microparticles repel water and begin regulating the water flow exclusively through the microchannels, effectively preventing passage through the foam's macrochannels[31]. This approach effectively blocks bulk water flow during evaporation process, minimizing the convective heat dispassion and sustaining a steady water supply. This advancement culminates in an average evaporation rate for the *p-SDWE* of 3.58 kg m$^{-2}$ h$^{-1}$, underscoring the efficacy of this innovative approach in enhancing the water evaporation process. In comparison, the *p-SE* demonstrated lower efficiency (2.43 kg m$^{-2}$ h$^{-1}$) in solar evaporation despite its rapid response (Fig. 6d). Thus, an efficient water management could be ideal in maintaining a high vapor generation.

Moreover, we examined the mechanism of enhanced water evaporation by *SDWE*s via the thin water evaporation. The thickness of the

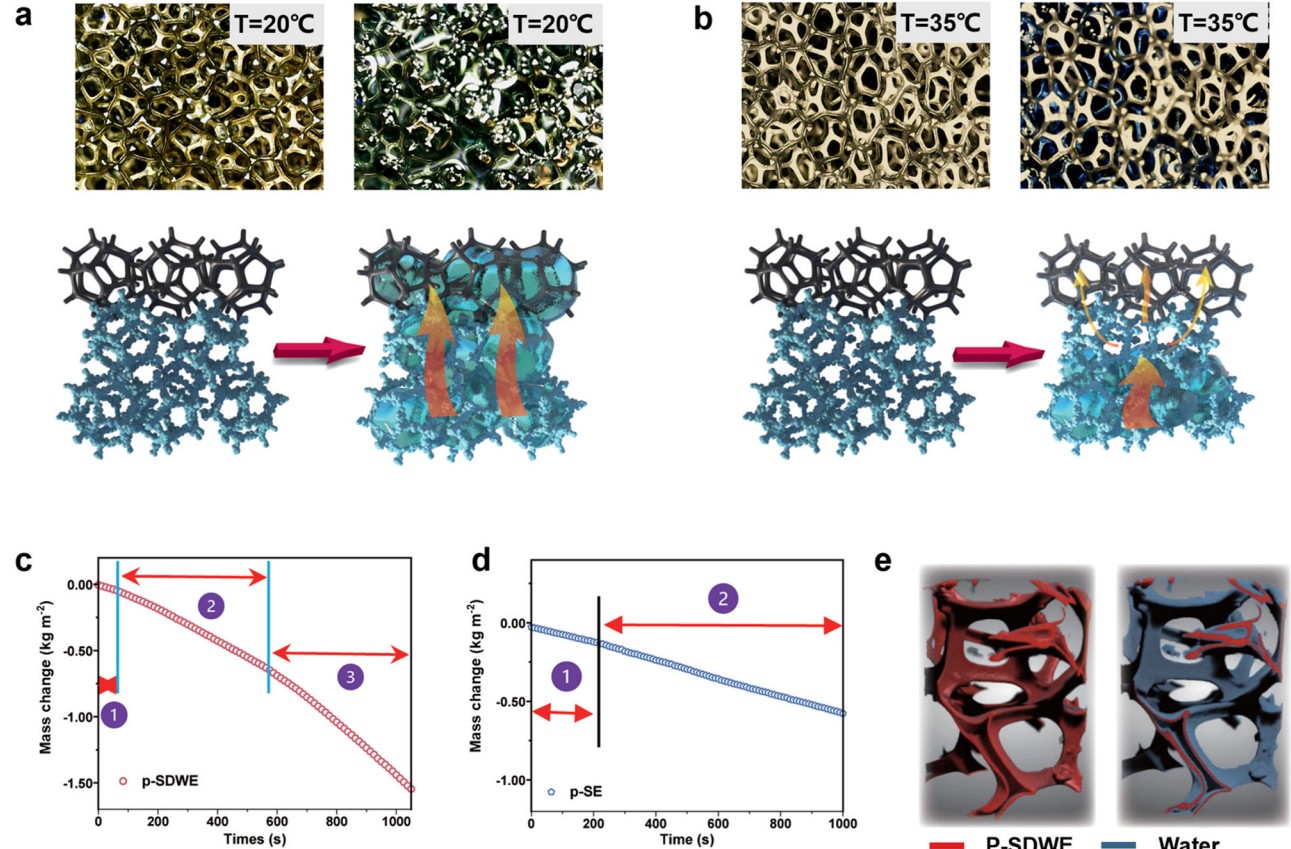

**Fig. 6 | Water pumping behaviors driven by different temperatures. a** the entire foam structure is filled with water at low temperatures (20 °C). **b** the bulk water is blocked by the superhydrophobic *PN10-g-SEC* layer, and it is pumped up to the *p-PDA* layer via the thin water transport channel of *p-PDA* at higher temperatures. **c** Mass change of water generated by *p-SDWE* under 1 sun irradiation. **d** Mass change of water generated by *p*-SE under 1 sun irradiation. **e** Thin water layer within *p-SDWE* measured by Micro-CT, where the resolution is 500 nm, 2000 projection.

water layer directly impacted the amount of thermal energy required for the liquid water-to-vapor transition during the water evaporation process. Micro-CT with a resolution of 500 nm was first utilized to quantify the thickness of water on the *SDWE*s[34]. The foam with the water supply systems was sealed and placed in the sample room, and the obtained 2D images and 3D reconstruction models at different positions were recorded. The KI solution was used as a contrast agent for better observation of the different phases[35]. As shown in Fig. 6e, the inner phase of red section was assigned to the sponge skeleton and the blue phase corresponds to the water phase. According to the CT scanning, no water filled the pores of the sponge itself, which further confirmed that the ultrathin water layer was generated within the whole sponge (Supplementary Movie 4)[36]. For the cross view of water phase, the calculated thickness of water ranged from 5.7 to 7.9 μm, and the average thickness was about 6.8 μm (Supplementary Fig. 16). The reduced thickness of water layer not only enhanced the water transport rate, but also impacted the evaporation process and its associated evaporation entropy. When water molecules were confined within a microchannel, they tended to escape from the surface as small clusters rather than individual molecules. As a result, the evaporation of water in this confined state led to a lower equivalent evaporation enthalpy (~1080 J g$^{-1}$) compared to the conventional latent heat of bulk water[37] (Supplementary Table 1). This characteristic of thin water layer transport within the foam could effectively mitigate the unnecessary heat required for vaporization[38].

Furthermore, we investigated the autonomous salt washing mechanism, simulating the operational conditions of *SDWE*s when light was obstructed by salt accumulation or during periods of

darkness when solar irradiation was unavailable. The evaporator was placed on a 3.5 wt% NaCl saline solution and subjected it to 5 sun illumination. This setup allowed us to observe both salt dissolution phenomena and temperature distribution on the *p-SDWE*'s top surface. As illustrated in Fig. 7a, salt crystals appeared on the *p-SDWE* surface after 6 h of illumination. The uneven temperature distribution, confirming the higher temperature on the salt crystals due to light absorption, hindered the photothermal effect and consequently reduced evaporation efficiency (Fig. 7b). To simulate a natural cycle, the light source was then deactivated and triggered the dissolution of the accumulated salt (Fig. 7c). Fig. 7d demonstrates a notable reduction in the surface temperature of the *p-SDWE* to 21.6 °C, aligning with the ambient temperature of the water container. This suggests an active transport of bulk water to the surface, followed by a convective backflow that effectively disperses the concentrated salt, simultaneously facilitating surface cooling[33,39]. This behavior suggests a transition in water flow to a dual-entrance channel system when the heat conversion falls below the LCST of the sporopollenin. Remarkably, within 40 min, all salt residues were reabsorbed into the solution, as the IR images confirmed a uniform surface temperature of 39.6 °C, indicative of a restored equilibrium (Fig. 7e, f). This efficient self-cleansing capability is integral to restarting the evaporation process, which demonstrates the potential of the *p-SDWE* design for sustainable long-term cycling processes.

As illustrated in Fig. 7g, h, comparative evaporation measurements were performed using 10 wt% salinity brine to evaluate water generation performance. Two reference samples were prepared for comparative analysis: one supplying thin water (denoted as *p-ShE*) and the other

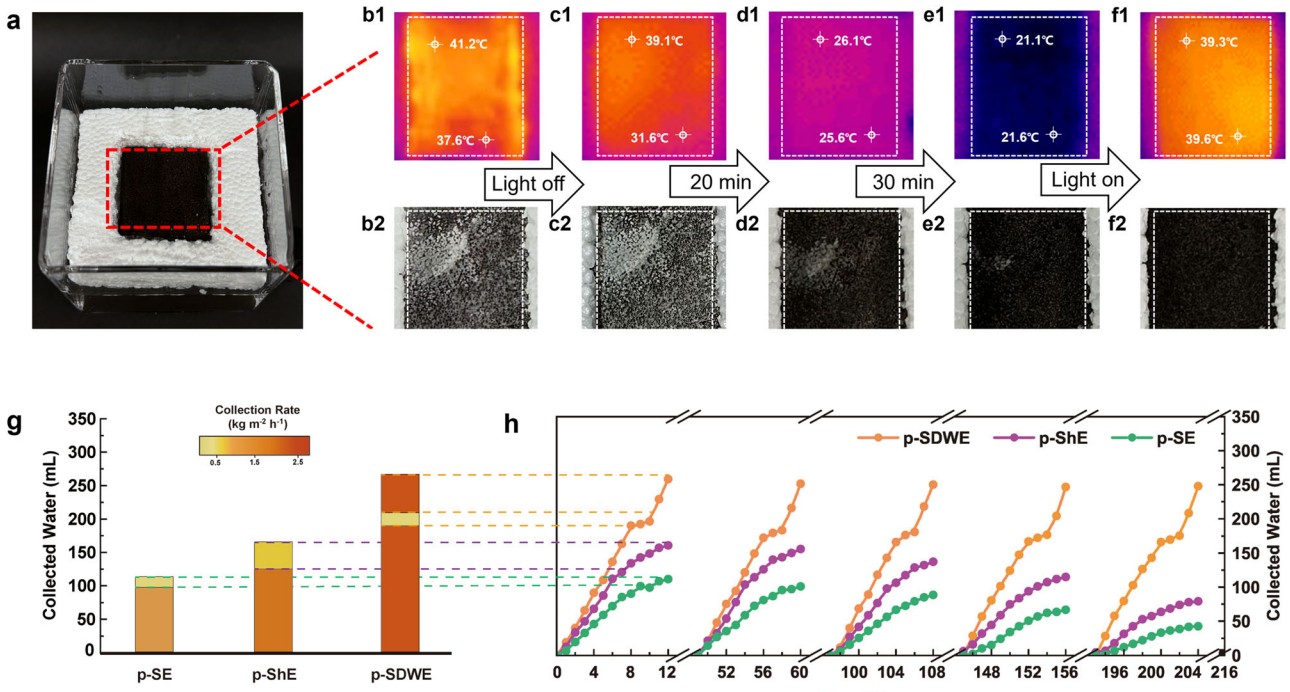

**Fig. 7 | Illustration of salt dissolving and backflow of *p-SDWE*. a** Experimental setup, Temperature distribution traced by IR Camera and top view digital images of the evaporation surface: **b1** and **b2** Salt crystallization. **c1** and **c2** Salt dissolution after simulator light deactivation, **d1** and **d2** Salt dissolution after 20 min. **e1** and **e2** Salt dissolution after 30 min. **f1** and **f2** Salt dissolution after 40 min. **g** The operational regime varying the collection rate during one cycle of *p-SDWE* compare with *p-ShE* and *p-SE*. **h** Long-term cycling performance using 10 wt% simulated seawater, where the reference sample of *p-ShE* denoting as the evaporator decorated with superhydrophobic SEC layer.

supplying bulk water (denoted as *p-SE*). The *p-SDWE* consistently out-performed these references, achieving a higher and more stable water collection rate of up to 2.45 kg m⁻² h⁻¹. Notably, the *p-SDWE* exhibited a distinct operational regime with the other two evaporators. *P-SDWE* initially generated water at a higher rate, but salt crystallization started to form on its surface after 8 h, interfering with heat conversion and localization. Subsequently, this salt accumulation triggered a self-washing phase, during which the collection rate significantly decreased as the system underwent the bulk water backflow process. After approximately two hours of washing, the *p-SDWE* resumed high-efficiency evaporation, effectively utilizing solar energy throughout the day.

In contrast, *p-ShE* demonstrated high initial evaporation rates (2.38 kg m⁻² h⁻¹) which occurred at an earlier salt accumulation (~ 7 h) due to its higher salinity and thinner water channels, significantly reducing its performance after salt accumulation, with recovery that was dependent on night washing. Conversely, *p-SE*, though delaying salt accumulation through convective flow within larger pores, incurred higher heat loss due to its cooling effect, resulting in lower water collection rates (1.26 kg m⁻² h⁻¹) and volumes. These observations demonstrated the superior performance of our dynamic water and thermal controlling system in *p-SDWE*, enhancing water evaporation rates and ensuring stable, long-term operation. This system markedly outperforms conventional evaporators, which rely solely on salt rejection or day-night washing.

### Laboratory and outdoor solar desalination
Thus, our *SDWE*s innovatively control the temperature fluctuations during the solar-driven water evaporation and functions as an adaptive solid gate for dynamic water-thermal controlling. It is noting worthy that a solar evaporator with controllable water management enables the continuous water supply during water-vapor phase transition while alleviating the accumulated salt, which are desirable for achieving higher evaporation rate.

To further evaluate the long-term water generation performance, the continuous solar evaporation performance of the *SDWE* was tested in a standard testing chamber for 6 h under one sun irradiation (Fig. 8a). The evaporation rate of *p-SDWE* remained constant at 3.58 kg m⁻² h⁻¹ and the water generation rate increased to 3.02 kg m⁻² h⁻¹, and remained constant with similar deposition and drainage characteristics that was similar to the observation at short-term test. Besides, the design produced an ultrathin water layer in the *SDWE* that contributed to achieving high energy efficiency. The energy efficiency($\eta$) was calculated using the following equation $\eta = mh_v/(C_{opt}P_0)$. Specifically, m represents the net evaporation rate and $h_v$ denotes the evaporation enthalpy of the water in the *SDWE*. The optical parameters assigned to the equation are referred to as $P_0$ for the solar irradiation power and $C_{opt}$ for the optical concentration (Supplementary Note 2)[40]. As shown in Fig. 8b, the *p-SDWE* achieved an energy efficiency of ≈93.9% under one sun, which was higher than the other two evaporators (*s-SDWE* and *r-SDWE*). This enhancement could be attributed to the incorporation of a switchable micro-channel, which minimized the heat loss to the water phase, resulting in only a 1.9% convection loss and a conduction heat loss of approximately 1.86% of the total received energy. The measured average reflection loss of *p-SDWE* over the broad solar spectrum was ~5.6%. Therefore, the sum of calculated energy losses was 13.0% under 1 sun illumination (Supplementary Note 3). Besides, the hydrophilic nature of the *p-PDA* layer reduced the heat transfer resistance at the interface, leading to improved heat transfer and facilitating the formation of a vaporized core. Thus, the *p-SDWE* showed a high performance in the solar-driven water evaporation.

The results confirmed that the surface would not succumb to flooding that could destroy the surface structure and reduce the water harvesting efficiency. The continuous solar desalination performance of the *SDWE* was also evaluated using simulated seawater with specific

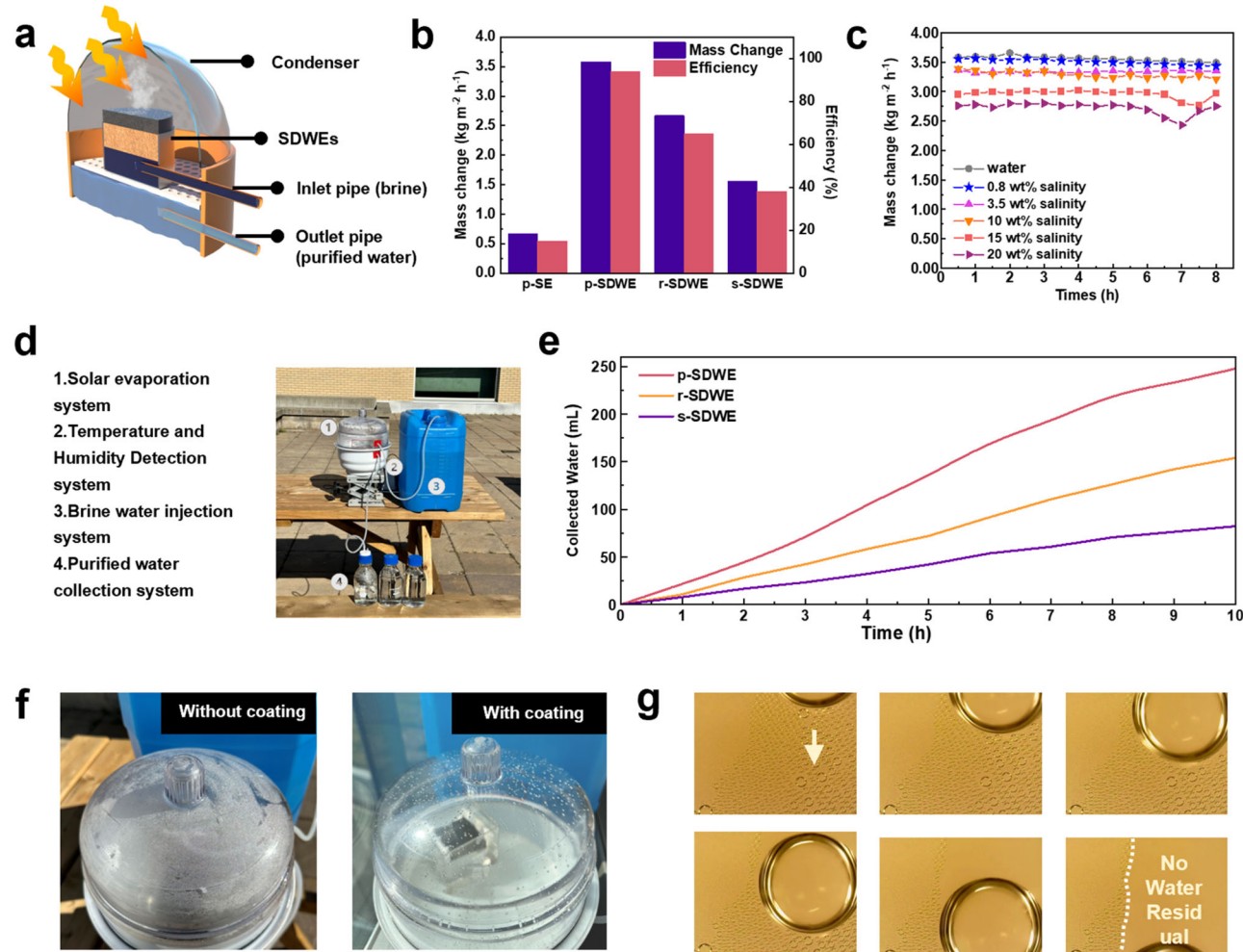

**Fig. 8 | Solar evaporation performance in an outdoor environment. a** Schematic illustration of solar-driven water evaporation. **b** Evaporation rate and efficiency generated by *SDWE*s. **c** Mass change of *p-SDWE* as a function of time for the evaporation of DI water, 0.8 wt%, 3.5 wt%, 10 wt%, 15 wt% and 20 wt% brine under one sun illumination. **d** A prototype solar water purification system simulating the practical water purification equipment. **e** The amount of purified water during 12 h of outdoor solar desalination. **f** Illustration of condenser applied in *SDWE* system with omniphobic liquid-like coating, compared with traditional device without liquid-repellent coating. **g** Droplet movement during evaporation process observed on the condenser's surface by microscope.

salinity (g of dissolved salt per kg seawater (wt%)), including Baltic sea (0.8 wt%), world ocean (3.5 wt%), and Dead sea (10 wt%), under one sun irradiation[41]. As shown in Fig. 8c, there was no visible salt crystal appeared on the top surface of the *p-SDWE* foam after the 8 h of continuous illumination due to the confined convection flow. Moreover, the average evaporation rate for Baltic seawater remained as high as 3.56 kg m$^{-2}$ h$^{-1}$, while for simulated seawater with salinities of world ocean water and Dead seawater, the rates corresponded to 3.34 kg m$^{-2}$ h$^{-1}$ and 3.22 kg m$^{-2}$ h$^{-1}$, respectively. When treating water with higher salinity, salt accumulation reduced the evaporation rate to lower levels after 6.5 h for 15 wt% saline water and after 6 h for 20 wt% saline water. Notably, after a single cycle of self-adaptive salt washing, which is depicted by the concave period on the mass change curve in Fig. 8c, the evaporation rates were restored to their original levels. After the solar desalination, the salinities of different simulated seawater samples decreased correspondingly (Supplementary Fig. 17a), which was far below the WHO and EPA drinking water standard. Moreover, the concentration of Na$^+$, Mg$^{2+}$, Ca$^{2+}$, and K$^+$ was reduced by orders of magnitude after solar desalination (Supplementary Fig. 17b). Even when the solar irradiation reached 5 kW m$^{-2}$, the occurrence of salt crystals could be resolved by the natural convection through the large pore size of the *SDWE* during the night or when impeded by accumulated salts (Fig. 7a). The introduction of flexible water

transport channel enables its recyclability and anti-salt-accumulation characteristic.

To demonstrate the possibility of continuous water generation, we conducted an outdoor experiment using the homemade solar evaporator system. A solar water purification prototype based on the optimized evaporator system was placed on the roof of the Engineering 3 at the University of Waterloo campus (Fig. 8d). The experiment was conducted from 08:00 to 20:00 under natural sunlight with an average solar heat flux of ~0.72 KW m$^{-2}$. In this setup, a *SDWE* sample with a diameter of ~10 × 10 cm and thickness of 1 cm was floated in a water container, and the container was insulated by a thermal insulation made of polystyrene foam to minimize heat loss. Condensed water formed on a transparent condenser, enhanced with an omniphobic liquid-like coating[42]. This coating imparts stability to the surface interface and reduces contact angle hysteresis, facilitating the seamless shedding of condensed water (Supplementary Fig. 18). Such an effective modification significantly augments the efficiency of the water harvester by ensuring unobstructed and clear light transmission (Fig. 8f), as the droplets slip off from the condenser's surface without water residual[43] (Fig. 8g). Additionally, an automatic water supply and pumping system were implemented using gravity, eliminating the need for external energy input. As shown in Fig. 8e, the *p-SDWE* could achieve an average water evaporation rate of 3.14 kg m$^{-2}$ h$^{-1}$ and water

purification rate of ~2.48 L m$^{-2}$ h$^{-1}$. Besides, the surface temperature of *SDWE* approached the peak temperature of around 50 °C in the thermal steady state. The temperature difference between *SDWE* and surrounding environment illustrated the localized heat supported by the photothermal layer of *PDA* and good thermal insulation of the SEC layer. These findings suggested that the *p-SDWE* evaporator offers a cost-effective solution in enhancing the performance of solar-driven water generation on a practical and large scale.

## Conclusions

We have successfully fabricated *SDWE*s with bilayer structure that enabled the continuous thin water supply and efficient thermal energy management. Taking advantage of the thermo-responsive layer, the water transport channel was switchable to allow for the passage of thin water within the inner microchannel. This water gating mechanism endowed the evaporator with a high solar-vapor conversion performance due to the thin water requiring low latent heat during evaporation compared to the bulk water. With the optimization of solar-driven water generation system, the solar evaporator could achieve a high vapor generation rate of up to 3.58 kg m$^{-2}$ h$^{-1}$ and 93.9% solar-to-vapor efficiency under 1 sun irradiation. Furthermore, the *SDWE* demonstrated its potential in solar-driven seawater desalination, contaminated water purification, and heavy metal ions removal. These findings provide fundamentals understandings to the water transport and phase transition at the evaporated interface, offering opportunities for the advancement of solar evaporator design.

## Methods

### Materials

Lycopodium clavatum pollen (obtained from Flinn Scientific Canada Inc.) was subjected to alkali extraction to obtain the sporopollenin shell for subsequent utilization. Nickel foam, N-isopropylacrylamide (97%), cerium (IV) ammonium nitrate (98.5%), potassium hydroxide (90%), phosphoric acid (85%), acetone (99.5%), ethanol (99%), ammonia solution (28–30%), Rhodamine B, potassium iodide (99%). All the chemicals were used as received from Sigma-Aldrich used without additional purification, unless stated otherwise. Milli-Q water (resistivity of 18.2 MΩ cm) was used to prepare the aqueous dispersions.

### Preparation of *PNm-g-SEC* microparticles

Briefly, we prepared sporopollenin (SEC) via alkaline extraction method and modified its outer surface with monomer of NIPAM via free radical polymerization[17,44]. We first treated the defatted pollen with 10 wt% potassium hydroxide (KOH) at 80 °C and stirred for 2 h to remove its internal cytoplasmic content. KOH-treated pollens were next subjected to acidolysis in 85% phosphoric acid (H$_3$PO$_4$) and then extensively washed with ethanol and water for further synthesis. Next, the surface modification of SEC microparticles was conducted under nitrogen environment, where cerium (IV) ammonium nitrate (CAN) (0.05 g 0.10 mmol) was added to the reaction flask forming radical sites on SEC through the reduction of ceric ions under magnetic stirring in an ice bath for 30 min for pre-synthesis. Next, NIPAM monomer (1.13 g, 10.0 mmol) was introduced to initiate the polymerization. Finally, the product was dialyzed against deionized water until the measured water conductivity remained constant. The as-prepared PNIPAM modified SEC microparticles designated as *PNm-g-SEC* where m corresponds to monomer mass changing from 5, 10, 20 mmol (initiator/monomer ratio-m is constant).

### Preparation of PDA@NiF foams

Firstly, the nickel foam (*NiF*) with dimensions of 5 cm × 5 cm × 10 mm was washed with acetone, ethanol and immersed in 1 M hydrochloric acid (HCl) for 1 h to remove the oxide layer. The treated *NiF* foam was then immersed in a 150 mL water/ethanol solution (v:v = 2:1) and ammonia aqueous solution (NH$_4$OH, 0.75 mL, 28–30%) was added

dropwise to the mixture under mild stirring at room temperature for 30 min. Dopamine hydrochloride (0.5 g) was dissolved in deionized water (10 mL) and injected into the above solution mixture. The color of this solution immediately turned to brown and gradually changed to dark brown. The reaction was allowed to proceed for 3 h. The foams coated with *PDA* layers (*p-PDA*) were obtained and washed with water three times to remove unreacted chemicals. As for the reference samples, *NiF* foam coated with smooth *PDA* layers (s-*PDA*) were prepared by changing the reaction conditions in the water-ammonia solution, while roughness *PDA* layer (r-*PDA*) was fabricated in water-ammonia solution at 55 °C.

### Fabrication of *SDWE* evaporators

The *SDWE* evaporators were fabricated using conventional di*p*-coating methods. *PDA@NiF* foams were immersed in specific depth of the *PNm-g-SEC* solution that was controlled by a variable speed motor-driven height positioning unit of DCAT 15. Then, the *PNm-g-SEC* microparticles were adhered to the foam and vacuum dried for the solar water evaporation test. This series of solar evaporator designated as *SDWE*s, while *s-SDWE*, *r-SDWE* and *p-SDWE* refers to the evaporator with the distinct structure of *PDA* layer, respectively.

### Solar water evaporation experiments

The water evaporation performance experiments were conducted in the laboratory using a solar simulator (AbetTech, M-LS Rev B) with a simulated solar flux of 1000 W m$^{-2}$ (1 sun). The samples were cut into cubes and mounted onto a polystyrene thermal insulating foam (surface area-5 × 5 cm; thickness-10 mm). The size of the samples was carefully aligned with the solar illumination and absorber surface, preventing additional light falling onto the non-sample region to avoid additional solar-thermal heating. Besides, the distance between the solar simulator and the evaporator was set to 20 cm. The water container along with the sample was placed on a sensitive electronic weighing balance (Radwag SMB-60/AS 60/220.R2) to measure the mass of the water as a function of time with a sampling rate of 1 data point per second. In this regard, the evaporation rate was quantified by correlating it with the mass loss/change ($\Delta m$) of water. Simultaneously, the temperature changes were recorded by mounting two K-type thermocouples at different positions ($z$), where one was mounted onto the top surface of the foam ($z = 10.0$ mm) and the second thermocouple was installed at the bottom of the foam ($z = 0$ mm). These thermocouples were connected to an electronic data logger for temperature monitoring, while thermal images of the absorber surface and bulk water were recorded using an infrared camera (FLIR TG167).

Before illuminating the setup, the evaporation rate under the dark condition was measured for 1 h and used as a reference for self-evaporation. Then, the loss in water mass, with and without the *p-SDWE*s, were recorded for the calculation of the evaporation rate. The solar-illuminated evaporation rate was determined by subtracting the dark-condition evaporation rate. All the evaporation rates were measured after stabilization under 1 sun for 10 min.

### Particle morphology and surface topography characterizations

The shape and structure of SECs were visualized and examined using scanning electron microscope (SEM) and supplementary optical microscopic images and videos captured using the Nikon LV ND microscope. The particle size was measured using a particle size analyzer (Anton Paar 1190) and a Nano Zetasizer (Malvern ZS90) with a temperature control system. The as-prepared SEC surface topography was characterized by a confocal laser microscope (Olympus LEXT OLS5000).

### Thermodynamic characterization

The thermal responsive characteristics was evaluated by performing turbidimetric measurements on the Varian (Cary 100 Bio) UV-vis

spectrometer equipped with a temperature controller and Micro-differential Scanning Calorimetry (DSC). The rheological properties of the concentrated suspension were characterized using Malvern Kinexus ultra+ rheometer with the cylindrical measuring system and a solvent trap to prevent water evaporation.

## Surface wettability characterization

The water static and dynamic contact angle measurements were conducted via sessile drop methods using the OCA 15 device (Dataphysics). For the measurement of the static contact angle, a 5 μL volume micro droplet was dispensed onto the test surface using an auto-dosing system equipped with a 500 μL needle. Side-view images or videos of the droplet behaviors were captured using a high-speed camera (IDS uEye camera), while the analysis system recorded the time-dependent contact angles and incorporating Image J analysis software for the results analysis.

The surface adhesion force measurements were conducted in a tensiometer with the accompanied a multi-axis sample stage. In particular, the adhesion force variation was measured by the force probe while approaching and retracting a 3 μL droplet from the surfaces at an approximate speed of 0.05 mm/s, while the droplet's size and shape were recorded by a high-speed camera.

## Liquid transport characterization

Aqueous liquids containing fluorescent dye (Rhodamine B, < 0.01 wt %) were introduced into the *SDWE* foam. Liquids were injected by driving the liquid through the capillary using a syringe pump with micromanipulator mounted on the confocal microscope. The fluorescent-dyed water was placed on one side of the foam and recorded water transport on specific area under time−series mode and z-stack mode.

The interfacial behaviors between water-air-solid surface were characterized by a confocal microscope (Zeiss LSM 510 Meta Laser Scanning Confocal Microscope) under the time−series frame mode and z-stack mode to distinguish the three phases. During the testing process, water-immersion objective lens was used with the observation magnification of 40×/1.3 water DIC.

## Data availability

The data supporting the findings of the study are included in the main text and supplementary information files. Raw data can be obtained from the corresponding author upon request. Source data are included in the source data file. Source Data file has been deposited in Figshare under accession code DOI link[45].

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

## Acknowledgements

K.C. Tam wishes to acknowledge the financial support from NSERC and CFI Canada. The work reported in this paper is part of the PhD thesis[46].

## Author contributions

Y.W. and W.Z. contributed equally to this work. Y.W., W.Z. and K.C.Tam conceived the research. Y.W., W.Z., Z.W. and K.C.Tam designed the research. Y.W., W.Z., Y.Lee and Y.Li developed the experimental setup and performed the experimental characterization. Y.W., W.Z., Z.W. and K.C.Tam interpreted the theoretical and experimental results. All the authors contributed to data reduction and data analysis. Y.W. and W.Z. wrote the manuscript with input from all authors.

## Competing interests

The authors declare no competing interests.
