## [Peer Review File · Nature Communications]

Thermo-Adaptive Interfacial Solar Evaporation Enhanced by Dynamic Water GatingEditorial Note: This manuscript has been previously reviewed at another journal that is not operating a transparent peer review scheme. This document only contains reviewer comments and rebuttal letters for versions considered at *Nature Communications*. Mentions of the other journal have been redacted.

REVIEWER COMMENTS

Reviewer #1 (Remarks to the Author):

The described bilayer-structured solar evaporator (SDWEs) employs a dual water transport system featuring a polydopamine (PDA) photothermal coating layer and a bottom thermo-responsive sporopollenin gating layer. While the design approach of integrating a water supply microchannel with a selectively permeable gating mechanism is conceptually intriguing, the work appears to lack significant innovation in both structural design and functional performance. Firstly, the use of PDA for the photothermal section is a well-established method in the literature, thus providing little novelty in terms of material selection. The core idea of having a switchable water gating layer based on the sporopollenin's temperature responsiveness isn't groundbreaking either; similar approaches have been explored in various thermal management systems. Moreover, the paper does not seem to offer any breakthroughs in performance enhancement or practical applicability. No substantial improvements in evaporation rates, energy conversion efficiency, or cost-effectiveness compared to existing systems are presented. Additionally, the actual real-world usability and long-term stability of such a system under varying environmental conditions remain unaddressed. Given the intense competition and rapid advancements in the field of solar evaporation technology, this work may struggle to contribute meaningfully to commercial and practical applications.

Reviewer #2 (Remarks to the Author):

Compared with the original manuscript, the revised manuscript has been improved. The innovation and importance of the work has been clearly highlighted, which can exhibit the differences and some advantages (treating high salinity brine) compared to the published paper I mentioned. Therefore, I think this manuscript may be suitable for Nature communications. However, I still have a few questions.

1. In Figure 7c, the concentration of the salt is wrong. For example, the salinity of seawater is about 3.5%. However, in Figure 7c, the salinity is 3.5‰.
2. In Figure 7c, why the mass change for 10% NaCl is much higher than that for 3.5 %? The authors should explain it.
3. Can the evaporator treat brine with higher salinity such as 15 wt% or even 20 wt%?

Reviewer #3 (Remarks to the Author):

The authors have addressed the concerns properly.

REVIEWER COMMENTS

Reviewer #1 (Remarks to the Author):

The described bilayer-structured solar evaporator (SDWEs) employs a dual water transport system featuring a polydopamine (PDA) photothermal coating layer and a bottom thermo-responsive sporopollenin gating layer. While the design approach of integrating a water supply microchannel with a selectively permeable gating mechanism is conceptually intriguing, the work appears to lack significant innovation in both structural design and functional performance. Firstly, the use of PDA for the photothermal section is a well-established method in the literature, thus providing little novelty in terms of material selection. The core idea of having a switchable water gating layer based on the sporopollenin's temperature responsiveness isn't groundbreaking either; similar approaches have been explored in various thermal management systems. Moreover, the paper does not seem to offer any breakthroughs in performance enhancement or practical applicability. No substantial improvements in evaporation rates, energy conversion efficiency, or cost-effectiveness compared to existing systems are presented. Additionally, the actual real-world usability and long-term stability of such a system under varying environmental conditions remain unaddressed. Given the intense competition and rapid advancements in the field of solar evaporation technology, this work may struggle to contribute meaningfully to commercial and practical applications.

Response: We thank the reviewer for his/her time in re-examining our manuscript and we appreciate the opportunity to further clarify the unique aspects of our research.

Firstly, we articulated and presented our comprehensive responses to the critical comments concerning the **novelty, performance, and significance** of our work. To further address the reviewer's concerns, we have provided a comprehensive response as described below.

1. The novelty of this research

It is important to highlight that the core innovation of this research is centered on the concept of **'dynamic water control'** - achieving thin water evaporation at high temperatures and an autonomous transitioning to a bulk water self-washing mode when surface temperature is impacted by presence of pollutants, such as sodium chloride crystals. This is realized by the bilayer-structured solar evaporator (SDWE) design which can successfully manage the fluidic flow within the dual water transport channel. To the best of our knowledge, this concept and design have never reported before.

1) Initially, our SDWE can implement a "water gating layer" in response to the high temperature during the operation of system. This gating layer can effectively impede bulk water saturation within the channel of the Ni foam, while at the same time delivering a thin layer of water through the porous PDA coating. Consequently, the reduces the thermal loss and optimize the utilization of solar energy for evaporation. Furthermore, this "dynamic water control" mechanism also plays another critical function by actively adapting the operation of the system to the external environmental conditions. When the accumulated salts reduce the system's photothermal efficiency, our SDWE structure's thermal-responsive layer shift from superhydrophobic to hydrophilic states, enhancing capillary action for effective pollutant removal through bulk water convection. Once the pollutants are eliminated, the temperature will further increase, triggering a transformation from hydrophilic back to superhydrophobic. This reinstates the thin water evaporation mode, thereby enabling a long-term, efficient, cyclic utilization of the solar evaporator.

2) The "dynamic water control" concept, propelled by thermal-responsive evaporation, introduces a stable, long-term cyclic operation mode, marking a novel approach that has not been previously documented in the field. Unlike traditional evaporators that depend on mere salt rejection or day-night cleaning cycles, our system offers an innovative concept that incorporates a self-cleaning mechanism. It capitalizes on temperature fluctuations caused by fouling or salt accumulation during solar-driven evaporation, thereby inducing an effective, autonomous salt self-washing process.

3) Moreover, we conducted a detailed study on the high-efficiency phenomena generated by this dynamic water control under high temperatures. We delved deeper into the distinctions in thin water transport under varying morphological conditions, thereby enriching our understanding in this field.

2. The motivation and the importance of this research

The motivation of this work stems from the challenge in the field of solar evaporation to actively and dynamically control water flow, balancing high performance with long-term operation sustainability. Recent advancements have attempted to address both **high efficiency and salt rejection**. Despite their efficiency and reduced salt fouling, current systems are encumbered by their rigid structures and passive operation cycles, limiting their long-term viability. Achieving autonomous solar-driven water evaporation, particularly in environments with high salt concentrations and organic pollutants, remains a formidable challenge.

To meet this demanding requirement, we designed the solar-driven water evaporation (SDWE) structure to achieve the aforementioned dynamic water control, capable of operating under practical conditions. As previously mentioned, the core innovation of this design is its proactive response and state transition in corresponding environmental conditions. During operation, the high interfacial temperature was induced and amplified by our designed pollen grafting that surpassed the Lower Critical Solution Temperature (LCST) of the polymer, achieving a water gating effect and thereby enabling thin water transport. This dynamic structure's capability effectively addresses long-term operational challenges in complex environments, such as those with high salinity or heavy organic pollutant, ensuring sustained and efficient functionality.

Furthermore, considering the practical utilization of this technology for freshwater generation, we've improved our solar evaporation device with an omniphobic liquid-like coating on the condensation surface, enhancing water collection rates and reducing the interference of the incident sunlight. Our integrated automatic water supply and pumping system further ensures a compact and self-sufficient clean water harvesting solution that is ideal for purifying high salinity water in various natural environments.

3. The impact and significance of this work.

A practical and sustainable interfacial solar evaporator must cater to diverse environmental challenges. This paper advances the solar evaporation field in three significant ways:

- 1) The results of this study, particularly the bilayer structure facilitates the dynamic water control, allowing the device to display flexible, active, and sensitive behavior in response to environmental changes. This dynamic water control opens a new research direction in the field of solar evaporation, drawing inspiration from the rapid responsiveness observed in robotics for tackling environmental challenges.
- 2) Moreover, the water collection structure reported in this paper enables the miniaturization and self-driving capabilities. It's adept at handling a wide range of complex aquatic environments, from not only the high salt contents but also high organic (i.e. bacteria) pollution environment.
- 3) The operating temperature environment can be modulated by introducing different polymers, thereby affecting the applicability of this dynamic water control. The introduction of organic substances shifts the operational range of LCST, accommodating a broader temperature spectrum. For instance, lower LCSTs are desirable in winter for effective dynamic water control, while higher LCSTs are required in summer. This flexibility significantly expands the research and application scope of solar evaporation.

4. Supplementary experiments

We have enriched our submission with supplementary experimental details to solidify our primary concept and findings. This includes emphasizing the structural design and elucidating the distinctive benefits and significance of applying our technology in the field of solar evaporation. Detailed discussions pertinent to these enhancements are provided below and can be found in the revised manuscript.

Furthermore, we've refined the manuscript's structure, ensuring a logical and coherent presentation of our design concept. The novelty and significance of our work are prominently featured in both the abstract and introduction sections, with the aim of fully conveying the essence of our dynamic water control innovation.

Comment “Firstly, the use of PDA for the photothermal section is a well-established method in the literature, thus providing little novelty in terms of material selection.”

Response: We thank the reviewer's careful examination of our work. In response to Reviewer 1's comments, we want to highlight and articulate the function of PDA. The selection of polydopamine (PDA) as the core material for constructing our evaporator is grounded on three key reasons:

1) At the core of our innovation is the dynamic control of water and heat. To maintain rigorous research standards, we opted for PDA, a well-established photothermal material. This choice allows us to focus on the system's design rather than expending resources on exploring alternative photothermal materials. Future work will delve into optimizing the photothermal layer, considering factors like cost, fabrication methods, and practicality. Additionally, PDA's inherent chemical structure and water affinity are advantageous for activating the water phase during evaporation.

2) The process of fabricating the PDA coating layer is highly controllable, with well-documented research on its thickness, particle size, and structural uniformity.

3) PDA's bio-inspired properties offer high adhesiveness, crucial for coating substrates. In our design, this adhesive quality is able to effectively be coating on the nickel foam and bonding with pollen particles. This structural design enables the dual-entrance water channel constructed, which is critical in facilitating dynamic water transport and further improving the evaporation efficiency and sustaining the continuous working regime.

In addition, to further highlight its functions and related physics for solar water evaporation, we have provided supplementary results and added new comments to elaborate on the solar-to-heat conversion and heat-to-water transfer. Detailed illustrations were included in revised version (page 12-13), as “ Additionally, by strategic design and control of the PDA nanoparticles' assembly, we enabled the photothermal layer with low light reflectance of p-PDA@NiF foams of ~5.6%.... showcasing its remarkable solar-to-heat conversion capability.” “Considering the increase in topographical texture with effective phonon scattering and low thermal diffusivity in PDA ($0.08 \text{ W m}^{-1} \text{ K}^{-1}$),... These findings imply that the existence of a thin water layer, enhanced by the surface characteristics of polymeric materials such as PDA, plays a significant role in boosting evaporation efficiency.”

Some good works and reviews on the photothermal materials have been included and summarized (*Ref.[2] Nano Energy 41 (2017) 269-284, Ref.[26] Physics Reports 981 (2022) 1-50. Ref.[4] Nature Nanotechnology, 13 (2018) 489-495*) (page 3), as “Recent advancements in solar evaporators have been driven by the development of materials with high photothermal efficiency², such as metal nanoparticles³, carbonaceous materials, semiconductors, and polymers. These advancements have led to significant improvements in evaporation efficiency made possible by the innovative design of evaporators with varied structural forms including 3D porous architectures⁴, 2D lamellae⁵, and 1D columnar structures.”

Comment “The core idea of having a switchable water gating layer based on the sporopollenin's temperature responsiveness isn't groundbreaking either; similar approaches have been explored in various thermal management systems.”

Response:

1) To further highlight the significant role and advantage of the thermo-responsive layer, we have expanded our discussion and conducted a comparative analysis with other relevant studies. We have conducted a comprehensive review of existing literature in this fields and analysis that encompasses various aspects, such as structural design, fluidic flow principles, cycling performance, resistance to salt fouling, and evaporation rate. Detailed results of the representative comparisons have been incorporated into Table S2, which indicated that the introduction of thermo-responsive layer to control the dynamic water supply have not been reported previously. Drawing on a thorough comparison and analysis of previous studies, we have incorporated these insights into the "Introduction" section, highlighting the novelty and significance of our thermo-responsive evaporation approach (page 3), as “Smart materials are catalyzing a transformative shift in robotics, paving the way for autonomous soft robots capable of self-adapting to environmental stimuli like heat, light, and

magnetism¹⁵. However, leveraging this inherent self-adaptability in solar evaporator is still nascent....The integration of these materials into solar evaporator domain will not only offer promising heightened sensitivity to environmental shifts but also ensures unmatched stability in system performance.”

Such innovative utilization of material properties and surface engineering in response to environmental stimuli provides new insights for sustainable water management strategies. That is to say, the foundational design principle of our work is distinctly different from previous approaches.

Table S2 A summary of reported evaporators, varying in structural design, fluidic flow principles, cycling performance, resistance to salt fouling, and evaporation rate.

Evaporator's structure	Fluidic flow	Salt removal method	Cycling operation time	Evaporation rate (kg m ⁻² h ⁻¹)	Solar evaporation efficiency	Net evaporation rate (kg m ⁻² h ⁻¹)	References
Multichannel porous hydrogel	Capillary driven flow	Salt crystal (2.5 wt%)	100h	1.64 (SD) 0.89 (SD) 0.89 (SD)	46.2% 33.3% 33.3%	1.12 (SD) 0.59 (SD) 0.59 (SD)	4
TEMPO-oxyl PDA-cellulose membrane	Unidirectional flow		227h	1.55	80.0%	1.262	5
2D Alkane coated with silver nanowire mesh	Unidirectional flow along the silver nanowire mesh	Open exposure	20 min	1.20		0.62	6
Translucent 2D-1D	Disordered	Light off mesh		2.34 (SD) 0.93 (SD) 0.93 (SD)		0.69 (SD) 0.30 (SD) 0.30 (SD)	7
3D hydrogel	Unidirectional pump to the top surface		20 days	3.2	96%	2.9	8
3D sponge	Unidirectional flow with large pore	Day-night switch		3.2	91.0%	1.4	9
3D porous foam with vertically oriented channel	Specific control water channel	Sub-irradiation	100h	2.40	99.4%	2.67	10
Multichannel conductive PV	Directional transport	Salt can be concentrated at specific sites		2.43		1.77	11
Janus surface silver sponge	Segmented water and vapor			2.21	80%	1.95	12
From particles and aligned water channels	Disordered flow	Sub-irradiation	100h	2.25	136.7%	1.14	13
Wickless channel water layer	Capillary flow	Salt crystal (2.5 wt%)		1.36	95%	1.16	14
Conductivity	Non-contact	No salt (0.5 wt%)		2.5	20%	2.3	15
Conical	Non-contact	Salt transport to specific area and recovery	49h	1.25	103.9%	1.25	16
Water evaporation surface with large pores and is not physically separated	Disordered flow	Salt concentrated at specific sites	20h	2.42	94.5%	2.12	17
Asymmetric porous and asymmetric areas	Multiphase flow	Salt concentrated at specific sites	547.10	2.45	90%	2.20	18
Thermo-responsive hydrogel	Unidirectional flow	Salt-free	10 ³ h ²	4.165		N/A	19
Asymmetric responsive (thickened) film	Unidirectional flow		100h	2.9 (2.0 wt%)	97.2%	N/A	20
Hybrid structure of porous evaporator and self-washing	Disordered flow with water evaporation	Self-washing	210 h	3.58	93.0%	3.33	This work

2) Furthermore, to underscore the benefits of incorporating a thermo-responsive layer into the solar evaporator, we have modified both the abstract and the introduction, restructured the main content, and included additional results and explanations in the manuscript. These revisions aim to articulate distinctly on the innovative aspects and significance of our work within the scientific community, clearly differentiating it from prior research.

We have clarified the working principles and mechanism from two aspects: achieving thin water evaporation at high temperatures and an autonomous transitioning to a bulk water self-washing mode in response to the surface temperature drops caused by light impeded by salt accumulation. This dual-role mechanism is integral for both regulating water supply and localizing heat, which is manifested in two ways: 1) Upon exposure to solar irradiation, as the temperature increases, sporopollenin displays hydrophobic properties and effectively seals off the larger inner channel. This redirection compels the water to navigate through a narrower PDA-lined microchannel. Additionally, the intrinsic hollow architecture of sporopollenin acts as a thermal barrier, minimizing heat loss to the bulk water and optimizing the energy retention. 2) Conversely, under reduced temperatures, such as salt accumulation that impedes the light absorption or at night when solar energy is unavailable, sporopollenin undergoes a transformation to a superhydrophilic state. This alteration activates the larger water transport channel, which efficiently expels salt accumulations by enhancing the transport of bulk water and harnessing convective forces. Such innovative utilization of material properties and surface engineering in response to environmental stimuli provides new insights for sustainable water management strategies. That is to say, the foundational design principle of our work is distinctly different from previous approaches.

Besides, we aim to clarify that our approach extends beyond merely regulating water into a thin layer for vapor generation. This functionality stems from our novel design of a dynamic water and thermal controlling evaporator, which integrates thermo-responsive sporopollenin. In this configuration, the sporopollenin, upon absorbing solar light, becomes hydrophobic at elevated temperatures, directing water through PDA microchannels, as detailed in Section 3.1. Beyond this, the sporopollenin can transition to a superhydrophilic state and its related function regimes, as explained in the Section 3.2. This transformation enables a dual-entrance water transport model, facilitating bulk water transport to the evaporator for salt backflow and washing, especially under conditions of light impediment due to salt accumulation or during night-time. Our objective is

to develop an interfacial solar evaporator endowed with dynamic water-thermal management capabilities. This design is crafted to operate at high efficiency and facilitate continuous, intelligent water generation, functioning autonomously without the need for manual intervention or additional energy inputs.

Comment “ Moreover, the paper does not seem to offer any breakthroughs in performance enhancement or practical applicability. No substantial improvements in evaporation rates, energy conversion efficiency, or cost-effectiveness compared to existing systems are presented. Additionally, the actual real-world usability and long-term stability of such a system under varying environmental conditions remain unaddressed.”

Response: We thank the reviewer’s time for reviewing our work and providing helpful comments. To address the reviewer’s concerns, we have clarified and expanded on the performance enhancement and practical applicability of our proposed system, focusing on two key aspects: technological performance and efficiency, and potential for scalability.

1. Technological Performance and Efficiency We have provided detailed analyses of net evaporation rates, water generation capabilities, as well as the effectiveness of salt washing and long-term cycling performance. Firstly, we have included a comprehensive comparison table (Table S2) in the manuscript that reviews existing literature in the field and analyzes various aspects to showcase our outstanding performance, including evaporation rate, structural design, fluidic flow principles, cycling performance, and resistance to salt fouling. In our tests, the proportion of natural water loss was minimal due to the 2D confined interfacial evaporation structure. As calculated and documented in the manuscript, the evaporation rate for our system was $3.58 \text{ kg m}^{-2} \text{ h}^{-1}$, with only $0.45 \text{ kg m}^{-2} \text{ h}^{-1}$ evaporation rate attributed to natural loss. Compared with the previous reported studies, the net evaporation rate of our paper is largely improved due to the concept and structure we utilized in the manuscript.

Moreover, we have conducted a comparative analysis of the long-term cycling and stability of our dynamic water control evaporation system. We have conducted a comparative analysis of its long-term efficiency against traditional salt washing evaporation systems, highlighting the new salt washing approaches induced by thermo-responsive evaporation, as demonstrated in **Figures 6**. Our design is engineered for high efficiency and enables continuous, intelligent water generation. It ensures a stable solar-driven water evaporation and collection process over 216 hours, functioning autonomously without the need for manual intervention or additional energy inputs. Unlike traditional evaporators that depend only on salt rejection or day-night cleaning cycles, the as-designed system incorporates a self-cleaning mechanism, where it capitalizes on temperature fluctuations caused by fouling or salt accumulation during solar-driven evaporation, to generate an autonomous salt self-washing process. Detailed discussion are described on page 17-18, as “Furthermore, we investigated the autonomous salt washing mechanism, simulating the operating conditions of SDWEs when light is obstructed by salt accumulation or during periods of darkness when solar irradiation is unavailable.... These observations demonstrated the superior performance of our dynamic water and thermal controlling system in p-SDWE, enhancing water evaporation rates and ensuring stable, long-term operation. This system markedly outperforms conventional evaporators, which rely solely on salt rejection or day-night washing.”

Figure 6 Illustration of Salt dissolving and backflow of p-SDWE under 5 sun illumination over 6 hours: **a**. Experimental setup, Temperature distribution traced by IR Camera and top view digital images of the evaporation surface: **b1** and **b2**. Salt crystallization, **c1** and **c2**. Salt dissolution after simulator light deactivation, **d1** and **d2**. Salt dissolution after 20 min, **e1** and **e2**. Salt dissolution after 30 min, **f1** and **f2**. Salt dissolution after 40 min. **g**. The operational regime varying the collection rate during one cycle of p-SDWE compare with p-SHE and p-SE, **h**. Long-term cycling performance using 10 wt% simulated seawater, where the reference sample of p-SHE denoting as the evaporator decorated with superhydrophobic SEC layer.

In addition, we have articulated the effectiveness of temperature-induced bulk water flow in removing salt, ensuring that the rate of salt backflow from the evaporator surpasses its rate of salt generation, thus ensuring the sustained utilization of SDWEs (**Figure S15** and **Supplementary Video S3**). The related illustrations were described in detail in the manuscript (page 15-16) as “Simultaneously, to assess water transport on a macroscale, we simulated real-world conditions encountered during solar-driven water evaporation. This involved characterizing two key processes: the supply of a thin water layer during vapor generation and the bulk water backflow during salt washing. ...To intuitively showcase the salt backflow capacity, we investigated the salt dissolution process by introducing 1 g of sodium chloride to the evaporator surface during vapor generation. The salt progressively dissolved, completely disappearing within two hours as shown in Figure S15a. Additionally, the bulk water transport to the top surface of the evaporator rapidly back flowed and drove the saturated brine into bottom water container under the action of a salinity gradient and gravity, as recorded in Figure S15b and Supplementary Video S3. These observations confirm the effectiveness of temperature-induced bulk water flow for removing salt, ensuring that the rate of salt backflow from the evaporator surpasses its rate of salt generation, thus ensuring the sustained utilization of SDWEs.”

Figure S15 a. Illustration of the salt redissolving procedure with additional 1gram sodium chloride of p-SDWE under 1KW m^{-2} illumination, **b**. Salt backflow driven by gradient and gravity

Hence, the 'dynamic water control' concept, driven by thermo-responsive evaporation, introduces a novel and sustainable operation mode. This mode enhances the evaporation rate, effectively manages salt accumulation,

and provides long-term benefits. In essence, our approach not only improves operational efficiency but also ensures the system's durability and consistent performance over time.

2. Potential for Scalability We have articulated the scalability of our technology, in terms of the system's adaptability, stability, cost-effectiveness, and sustainability, underscoring its suitability for real-world applications. In the abstract, we emphasized the key issue that impedes the widespread adoption of the current technology, which also served as one of the motivations for our work (page 3) as "Scaling this technology for seawater treatment has encountered significant challenges, ...attaining autonomous solar-driven water evaporation continues to be a formidable challenge."

With regards to adaptability, we have highlighted in the manuscript that the "dynamic water control" concept played a crucial role: actively adapting to external environmental conditions. Specifically, our SDWE is capable of employing a "water gating layer" in response to the high temperature during the operation of the system. This gating layer can effectively impede bulk water saturated within the channel created by Ni foam, while provide the thin water supply mode by porous PDA coating. Consequently, reducing thermal loss and preventing the insufficient utilization of solar energy. When the accumulated salts reduce the system's photothermal efficiency, our SDWE structure's thermal-responsive layer shift from superhydrophobic to hydrophilic states, enhancing capillary action for effective pollutant removal through bulk water convection. Once the pollutants are eliminated, the temperature will further increase, triggering a transformation from hydrophilic back to superhydrophobic. This reinstates the thin water evaporation mode, thereby enabling its adaptability in real application.

For further improvement and optimization of the proposed solar evaporator in different seasons or regions, we could easily fine-tune the phase transition temperature of the thermo-responsive layer through copolymerization. That is, the gating function could work in different temperatures, and not being limited by the surrounding temperature. Therefore, this design concept can be readily expanded to our proposed solar evaporation system with dynamic water control. In future, we aim to fabricate materials that endow our Solar Driven Water Evaporators (SDWEs) with increased adaptability and flexibility for diverse working conditions. We recently published a paper on the preparation of a scalable substrate for controlling wettability transitions (JACS Au). Similar thermo-responsive materials could be applied in this study, allowing for dynamic water control and evaporation under different temperature conditions.

Secondly, we have provided a comprehensive examination of the controllable management of fluidic flow within the evaporator, its superior evaporation performance, and the benefits of self-adaptive salt washing in Section 3.1 and 3.2 of our manuscript. These sections underscore the system's robust capacity to efficiently process high salinity seawater and adapt to variable temperature conditions. Furthermore, we have detailed the real-world outdoor applications in Section 3.3, further reinforcing the practical viability and the strong potential for scale-up and maintaining long-term stability in practical applications.

We acknowledge that the cost-effectiveness of the evaporation system is a crucial factor in promoting the technology for practical use. Cost considerations for the solar evaporator encompass both the design and use of the base materials, as well as long-term maintenance and infrastructure expenses. Firstly, we utilize sporopollenin materials sourced from nature rather than synthetic alternatives for constructing the evaporator. This choice not only reduces costs but also enhances environmental sustainability. The photothermal materials could be selected based on cost-effectiveness, further reducing the production cost. Moreover, we are exploring the replacement of nickel foam with carbon foam, which aligns with our commitment to using sustainable and carbon-neutral materials, significantly advancing the design and construction of the solar evaporator beyond existing concepts.

Our approach integrated sustainability considerations into the design of the solar evaporator, ensuring its long-term operational viability. Compared to traditional systems, our proposed technology significantly reduces the maintenance costs. Additionally, we aim to enhance the functionality and efficiency of our solar evaporation device by applying an omniphobic liquid-like coating on the condensation surface (page 21-22) as "Condensed

water forms on a transparent condenser, enhanced with an omniphobic liquid-like coating. This coating imparts stability to the surface interface and reduces contact angle hysteresis, facilitating the seamless shedding of condensed water (Figure S18). Such an effective modification significantly augments the efficiency of the water harvester by ensuring unobstructed and clear light transmission (Figure 7f), as the droplet slip off from the condenser's surface without residual water (Figure 7g).” This innovation not only boosts water collection rates but also minimizes the interference of light, thereby optimizing the effective water collection rate. Furthermore, our design is uniquely coupled with self-adaptive features that utilize the energy transformations occurring within the system, eliminating the need for external energy inputs. This energy-efficient strategy promoted the practical application of our technology in freshwater generation, demonstrating a significant advancement in solar evaporation technology.

Comment “Given the intense competition and rapid advancements in the field of solar evaporation technology, this work may struggle to contribute meaningfully to commercial and practical applications.”

Response: We wish to reiterate the societal and academic benefits of our work. Our design introduces a novel concept and mechanism in the field of solar evaporation, allowing for an autonomous transition between high-performance evaporation and effective self-cleaning modes. This approach not only addresses the challenge of salt accumulation but also aligns with the principles of sustainable solar evaporator design, offering a new direction and thought process for future developments.

We aim to highlight key aspects that underscore the relevance and potential impact of our work, making it a suitable contribution for publication in Nature Communications:

1. To the best of our knowledge, this is the first report of such a design. Our design introduces a novel concept and mechanism in the field of solar evaporation, allowing for an autonomous transition between high-performance evaporation and effective self-cleaning modes. The introduction of dynamic water control in our design is not merely an innovation in solar evaporation; it's akin to the rapid adaptability of robotics, poised to effectively address environmental challenges. Moreover, the multi disciplines, such as polymer science, surface engineering, and fluid dynamics broadens the appeal of our work. It brings fresh perspectives and methodologies, fostering cross-disciplinary collaboration and pushing the boundaries of research in sustainable water and energy harvesting.
2. Our work revolutionizes the standard approach by moving beyond the conventional rigid evaporator designs. We present a groundbreaking concept and design that endows the evaporator with the capability to autonomously switch between high-performance evaporation and effective cleaning modes. This adaptability equips the device with flexible, active, and responsive behavior, demonstrating the potential to propel the evolution of the next-generation solar-driven evaporation technologies.
3. A robust, scalable and sustainable technology with potential for off-grid application and compatibility with existing water treatment and solar energy systems, setting a new standard for sustainable water harvesting and generation.

Reviewer #2 (Remarks to the Author):

Compared with the original manuscript, the revised manuscript has been improved. The innovation and importance of the work has been clearly highlighted, which can exhibit the differences and some advantages (treating high salinity brine) compared to the published paper I mentioned. Therefore, I think this manuscript may be suitable for Nature communications. However, I still have a few questions.

Response: We thank the referee for reviewing our paper and offering the positive comments. We are pleased that the referee appreciates our effort to improve the manuscript.

1. In Figure 7c, the concentration of the salt is wrong. For example, the salinity of seawater is about 3.5%. However, in Figure 7c, the salinity is 3.5‰.

Response: We thank the reviewer for this helpful suggestion. We have corrected the mistake and modified the label in figure and the text.

2. In Figure 7c, why the mass change for 10% NaCl is much higher than that for 3.5 %? The authors should explain it.

Response: We thank the reviewer for highlighting this point. We have corrected the mismatch between the label and the corresponding line in the figure. As the right value was claimed in the manuscript (page 21) “As shown in **Figure 7c**, there was no visible salt crystal appeared on the top surface of the p-SDWE foam after the 8 h continuous illumination due to the confined convection flow. Moreover, the average evaporation rate for seawater (0.8 wt%) remained as high as $3.56 \text{ kg m}^{-2} \text{ h}^{-1}$,...”

Additionally, we have conducted further experiments on the treatment of brine with higher salinity levels, such as 15 wt% and 20 wt%, and have included the related data in Figure 7c. We have added the related discussion in page 21 “...while for simulated seawater with salinities of 3.5 wt% and 10 wt%, the rates corresponded to $3.34 \text{ kg m}^{-2} \text{ h}^{-1}$ and $3.22 \text{ kg m}^{-2} \text{ h}^{-1}$, respectively. When treating water with higher salinity, salt accumulation reduced the evaporation rate to lower levels after 6.5 hours for 15 wt% saline water and after 6 hours for 20 wt% saline water. Notably, after a single cycle of self-adaptive salt washing, which is depicted by the concave period on the mass change curve in Figure 7c, the evaporation rates were restored to their original levels.

Furthermore, we have thoroughly reviewed and standardized the labels, units, and abbreviations throughout the figures and manuscript

Figure 7 a. Schematic illustration of solar-driven water evaporation, b. Evaporation rate and efficiency generated by SDWEs, c. Mass change of p-SDWE as a function of time for the evaporation of DI water, 0.8 wt%, 3.5 wt%,

10 wt%, 15 wt% and 20 wt% brine under one sun illumination, d. A prototype solar water purification system simulating the practical water purification equipment, e. The amount of purified water during 12 h of outdoor solar desalination, f. Illustration of condenser applied in SDWE system with omniphobic liquid-like coating, compared with traditional device without liquid-repellent coating, g. Droplet movement during evaporation process observed on the condenser's surface by microscope.

3. Can the evaporator treat brine with higher salinity such as 15 wt% or even 20 wt%?

Response: Thank you for your insightful feedback. We conducted additional experiments on the water treatment of brine with higher salinity levels, specifically 15 wt% and 20 wt%. The corresponding data are presented in Figure 7c. Through measurements of mass change during desalination, the results demonstrated that our evaporators were capable of treating brine with high salinity and could also proceed the salt washing process. As shown in Figure 7c, a concave period occurred, which was attributed to a reduction in the surface temperature arising from salt accumulation and staging during the salt washing process. After a specific time period—corresponding to 1 hour for 15 wt% saline water and 1.5 hours for 20 wt% saline water—the evaporation rate returned to its previous high level. Moreover, we have incorporated a new discussion into the manuscript. Please refer to the revised section in response to comment 2.

Reviewer #3 (Remarks to the Author):

The authors have addressed the concerns properly.

Response: We thank the referee for the time spent on our paper and complimentary comments. We are pleased that the referee appreciates our effort to improve the manuscript.

REVIEWERS' COMMENTS

Reviewer #1 (Remarks to the Author):

The concerns have been appropriately addressed by the authors.

Reviewer #2 (Remarks to the Author):

After much consideration, I think this manuscript may be more suitable for [REDACTED], rather than Nature Communications, since the key novelty “dynamic water control” is needless for outdoor practical application.

The reasons are listed below:

(1) Just as the author says, the key novelty of this manuscript is the “dynamic water control”. In fact, I had been almost persuaded during the last reply. However, the new data in Figure 7c in the reply lets me feel that the key novelty “dynamic water control” is needless. As shown in Figure 7c, the evaporator can maintain stable and high evaporation rate during 6h irradiation under one sun. In this experiment, the total solar irradiation is 6 kW m^{-2} , which is much higher than the US annual average daily solar irradiation ($\sim 4.5 \text{ kW m}^{-2}$). It means that the evaporator can maintain high evaporation rate during the whole day irradiation (outdoor), and then the evaporator can realize self-cleaning during nights without the need of extra “dynamic water control”. It means that the “dynamic water control” that realized by additional reagents (thermal responsive material) and materials (relative expensive Ni form) and corresponding complex modification process are not necessary. Thus, from this perspective, this manuscript may struggle to contribute meaningfully to commercial and practical applications. It is the main reason for me to change my mind.

(2) Another subordinate reason is that there are abundant mistakes or negligence in this manuscript, which I have proposed in last review. In fact, in this reply, there is also some spelling mistake, which makes me think the authors are not seriously.

As the declared novelty “dynamic water control” of this manuscript is not necessary for practical application, the significance of this manuscript is decimated, and this manuscript is not suitable for Nature Communications. However, considering the improvement of this manuscript, I recommend its acceptance on [REDACTED].

REVIEWER COMMENTS

Reviewer #2 (Remarks to the Author):

After much consideration, I think this manuscript may be more suitable for [REDACTED], rather than Nature Communications, since the key novelty “dynamic water control” is needless for outdoor practical application.

Response: We thank the referee for providing additional comments on our paper. To address this reviewer’s concerns, we have outlined our responses to the specific comments, especially on the significance of our work.

The reasons are listed below:

(1) Just as the author says, the key novelty of this manuscript is the “dynamic water control”. In fact, I had been almost persuaded during the last reply. However, the new data in Figure 7c in the reply lets me feel that the key novelty “dynamic water control” is needless. As shown in Figure 7c, the evaporator can maintain stable and high evaporation rate during 6h irradiation under one sun. In this experiment, the total solar irradiation is 6 kW m^{-2} , which is much higher than the US annual average daily solar irradiation ($\sim 4.5 \text{ kW m}^{-2}$). It means that the evaporator can maintain high evaporation rate during the whole day irradiation (outdoor), and then the evaporator can realize self-cleaning during nights without the need of extra “dynamic water control”. It means that the “dynamic water control” that realized by additional reagents (thermal responsive material) and materials (relative expensive Ni form) and corresponding complex modification process are not necessary. Thus, from this perspective, this manuscript may struggle to contribute meaningfully to commercial and practical applications. It is the main reason for me to change my mind.

Response: We thank the reviewer for the careful examination of our work. In response to Reviewer 2’s comments, we want to highlight and articulate the novelty and significance of the “dynamic water control” described in our paper.

Firstly, as shown in Figure 7c, the evaporator could maintain a stable and high evaporation rate during 6 hours of irradiation under one sun due to the use of a general photothermal material, PDA, which has a constrained photothermal efficiency. To further improve the evaporation rate in future solar-driven evaporation technology, it is essential to utilize highly efficient photothermal materials to enhance heat transfer for evaporation. However, this improvement leads to a serious issue on the rapid salt accumulation. For example, we are currently working on highly efficient photothermal nanomaterials as a coating layer to drive evaporation. The increased evaporation rate results in rapid salt accumulation within 2 to 3 hours of solar irradiation. This issue limits the overall evaporation performance during long-term operation and is not adequately addressed by the general day-night washing system. Therefore, the design of a "dynamic water control" system is gaining attention as a means to further improve evaporation efficiency. Additionally, the salt blocking issue has become a major obstacle in current technological development, limiting performance and practical applications, as substantiated by numerous reported studies. ([1] Nature, 2023, 622(7983): 499-506, [2] Advanced Materials, 2023, 35(24): 2301596, [3] Nature Communications, 2022, 13(1): 849, [4] Desalination, 2023, 567: 116993.)

Secondly, solar-driven water evaporation technology shows potential in various applications utilizing solar energy, such as high-salinity wastewater treatment to achieve zero-liquid disposal. This process involves applying constant irradiation flux and extremely high salinity water, which can lead to the rapid accumulation of salt, eventually blocking the light absorption area and water transport channels. By integrating the design of “dynamic water control,” high-salinity wastewater treatment can be improved to achieve long-term and stable evaporation cycling performance.

Therefore, our design not only enhances evaporation efficiency but also provides a design framework enabling the development and adaption of this technology for many types of multidisciplinary applications.

(2) Another subordinate reason is that there are abundant mistakes or negligence in this manuscript, which I have proposed in last review. In fact, in this reply, there is also some spelling mistake, which makes me think the authors are not seriously.

As the declared novelty “dynamic water control” of this manuscript is not necessary for practical application, the significance of this manuscript is decimated, and this manuscript is not suitable for Nature Communications. However, considering the improvement of this manuscript, I recommend its acceptance on [REDACTED].

Response: Thanks for your careful review and comments. We have checked and corrected the mistakes in the manuscript.